# AI-driven projection tomography with multicore fibre-optic cell rotation

Jiawei Sun [1,2,3] ✉, Bin Yang[3], Nektarios Koukourakis[2,3], Jochen Guck [4] & Juergen W. Czarske [2,3,5,6] ✉

Optical tomography has emerged as a non-invasive imaging method, providing three-dimensional insights into subcellular structures and thereby enabling a deeper understanding of cellular functions, interactions, and processes. Conventional optical tomography methods are constrained by a limited illumination scanning range, leading to anisotropic resolution and incomplete imaging of cellular structures. To overcome this problem, we employ a compact multi-core fibre-optic cell rotator system that facilitates precise optical manipulation of cells within a microfluidic chip, achieving full-angle projection tomography with isotropic resolution. Moreover, we demonstrate an AI-driven tomographic reconstruction workflow, which can be a paradigm shift from conventional computational methods, often demanding manual processing, to a fully autonomous process. The performance of the proposed cell rotation tomography approach is validated through the three-dimensional reconstruction of cell phantoms and HL60 human cancer cells. The versatility of this learning-based tomographic reconstruction workflow paves the way for its broad application across diverse tomographic imaging modalities, including but not limited to flow cytometry tomography and acoustic rotation tomography. Therefore, this AI-driven approach can propel advancements in cell biology, aiding in the inception of pioneering therapeutics, and augmenting early-stage cancer diagnostics.

Optical tomography has ascended as an emerging label-free microscopic technique that captures intricate, three-dimensional (3D) subcellular structures. This paradigm-shifting modality has redefined how researchers decipher cellular processes, unravel disease mechanisms, and evaluate treatment responses, thereby pushing the frontiers of biomedical exploration[1–4]. Conventional optical cell tomography typically relies on illumination scanning to obtain projections at various orientations, yielding significant resolution improvement in microscopy[4–6]. Furthermore, therapeutic evaluation of targeted drug treatment at the single-cell level becomes feasible through optical cell tomography[7]. Nevertheless, the finite numerical aperture of microscope objectives constrains the illumination scanning angle coverage, resulting in the axial resolution of the illumination scanning tomography being inferior to the lateral resolution. This causes a fairly large blank area along the optical axis in the spatial domain of the tomographic reconstruction, known as the missing cone problem[8]. Iterative reconstruction algorithms[8,9] such as filtered back-projection combined with analytic continuation approach[10] and edge-preserving regularization[11] are developed to extrapolate the missing information in the limited-angle tomography. Recent advances in deep learning-

[1]Shanghai Artificial Intelligence Laboratory, Longwen Road 129, Xuhui District, 200232 Shanghai, China. [2]Competence Center for Biomedical Computational Laser Systems (BIOLAS), TU Dresden, Helmholtzstrasse 18, 01069 Dresden, Germany. [3]Laboratory of Measurement and Sensor System Technique (MST), TU Dresden, Dresden, Germany. [4]Max Planck Institute for the Science of Light & Max Planck-Zentrum für Physik und Medizin, 91058 Erlangen, Germany. [5]Cluster of Excellence Physics of Life, TU Dresden, Dresden, Germany. [6]Institute of Applied Physics, TU Dresden, Dresden, Germany. ✉e-mail: sunjiawei1@pjlab.org.cn; juergen.czarske@tu-dresden.de

based limited-angle tomography further optimize the computation efficiency of the reconstruction process[12–14]. Nevertheless, these reconstruction approaches are mostly based on approximation algorithms or physics-informed models, a full-angle tomographic scan of the cell remains necessary to provide the prior knowledge for validating the reconstructed images.

Numerous cell rotation strategies have been advanced to facilitate full-angle optical tomography of cells. Straightforward methods involving the mechanical rotation of cells have been explored for full-angle optical tomography. However, complex sample preparation, such as fixing the cell position using a micro-tube[15] or a fiber tip[16] is required. Contactless sample rotation approaches empowered by microflow[17–20], dielectrophoretic field[21,22] or acoustic microstreaming[23–25] simplify sample preparation for full-angle optical tomography while minimizing cell damage and preserving sample integrity. Furthermore, optical manipulation uniquely offers precise and accurate control over cell rotation, enabling targeted rotation of cells within the 3D space[26]. This level of control is essential to ensure the quality and reliability of tomographic reconstruction[27]. Fiber-optic manipulation employs optical fibers to deliver light and generate optical forces, providing remote and non-invasive control of cell rotation on a microfluid chip[28–30]. By decoupling the manipulation and imaging axes, these fiber-optic manipulation systems can be adapted to different microscopy setups without interfering with the tomographic imaging process. We previously reported a multicore fiber (MCF) optical manipulation system that enables stable and contactless cell rotation around all

three axes within a fiber-optic trap[30]. Nevertheless, realizing tomography based on fiber-optic manipulation remains a substantial challenge, primarily due to the absence of a gold-standard method for accurately measuring the cell rotation angle. The accuracy of the tomographic reconstruction is intrinsically tied to the correct determination of the cell's rotation angle derived from two-dimensional (2D) projections, which continues to be a significant challenge in this field. Additionally, the typical reconstruction procedure for cell-rotation tomography demands complex and computationally intensive methods for pre-processing 2D projections acquired at different rotation angles in order to generate high-resolution 3D images of cells[18,19]. Consequently, online pre-processing of these projections continues to pose significant difficulties. Recently, the development of artificial intelligence (AI) and computer vision has revolutionized various aspects of optical metrology[31,32]. These advancements have led to paradigm shifts in microscopy[33–35], including super-resolution[36,37], cell segmentation[38], and virtual staining[39,40]. Despite considerable advancements in computer vision technology, its integration into optical cell-rotation tomography has yet to be extensively investigated.

In this paper, we introduce an AI-driven optical projection tomography (OPT) system that utilizes the multi-core fiber-optic cell rotator (MCF-OCR). This innovation effectively bridges the existing gap between fiber-optic manipulation and optical tomography. The core of this system is an AI-driven autonomous tomography reconstruction workflow powered by emerging computer vision technologies, enhancing the robustness and efficiency of optical tomography

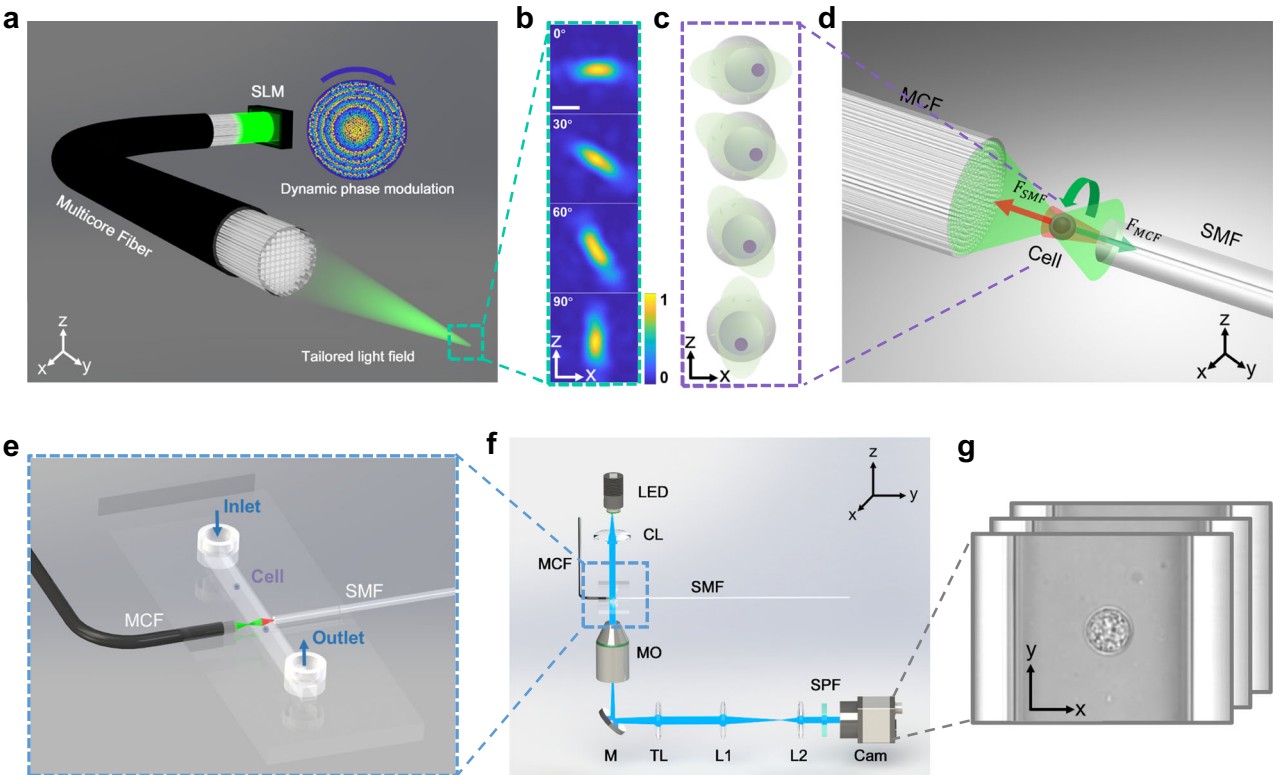

**Fig. 1 | Principle of the multicore fiber-optic cell rotation (MCF-OCR) powered optical projection tomography system. a** Spatial light modulator displaying computer-generated holograms for tailored light field generation through the MCF. **b** Cross-sectional view of the rotating elliptical beam in the optical manipulation region. **c** Cells are trapped in the MCF-OCR as the scattering forces from the two laser beams counterbalance each other. The trapped cells follow the rotation of the elliptical beam driven by the gradient forces generated by the heterogeneous internal refractive index distribution. **d** The MCF-OCR system comprises an MCF

and an opposing single-mode fiber to enable effective cell rotation and manipulation. **e** Cell delivery to the optical manipulation region through microflow.
**f** Simplified experimental setup of the optical projection tomography system. LED light-emitting diode, CL condenser lens, SMF single-mode fiber, MO microscope objective, M mirror, TL tube lens, L1, L2 achromatic lens, SPF short-pass filter, Cam camera. **g** Microscopic video recording optically controlled cell rotation for subsequent tomographic reconstruction.

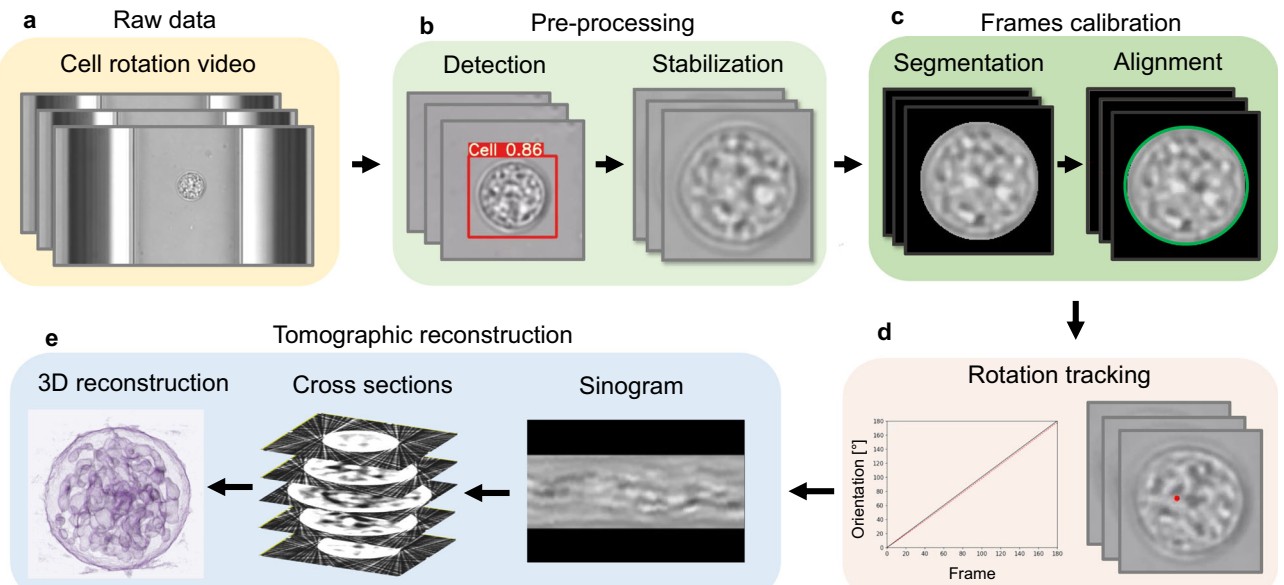

**Fig. 2 | Autonomous tomographic reconstruction workflow. a** Full-field optical microscope images show cell rotation facilitated by the MCF-OCR within a micro-fluidic channel. **b** An object detection convolutional neural network (CNN) autonomously identifies cells within the microscope images, cropping the detected region for stabilization via a dedicated algorithm. **c** The cell is separated from the background using an image segmentation deep neural network (DNN), after which the images are meticulously aligned. **d** Optical flow facilitates tracking and quantifying the rotation angle. **e** Sinograms are generated from the pre-processed projections and the corresponding rotation angle, enabling 3D intensity distribution reconstruction.

systems. The workflow involves object detection convolutional neural network (CNN) for real-time pre-processing of the projections, while deep learning is employed for cell segmentation from the background, significantly improving the quality of 3D reconstruction and enabling potential implementation for cell position alignment across frames. The Harris corner detector is utilized to extract features for rotation tracking, and the precise rotation angle of the cell is determined using the optical flow method. With the accurate rotation angle and projections, the 3D intensity distribution of the cell can be reconstructed using the inverse Radon transform. To validate the performance of our proposed autonomous tomographic reconstruction workflow, a cell phantom is implemented to characterize the accuracy compared to conventional optical tomography approaches. Moreover, we experimentally validate the proposed workflow by reconstructing HL60 human leukemia cancer cells, rotated using the MCF-OCR device. Our results indicate that this autonomous computational workflow can potentially serve as a generalized OPT workflow, thereby facilitating greater accessibility to full-angle tomography approaches for a wide range of biomedical research focusing on single-cell analyses.

## Results

### Projection tomography using a multi-core fiber-optic cell rotator (MCF-OCR)

The stability of cell rotation is a crucial element for achieving optimum cell tomographic reconstruction. To this end, as depicted in Fig. 1, we introduce the MCF-OCR-powered tomography system, consisting of an MCF and an opposing single-mode fiber. This innovative system employs a dynamically controlled rotating elliptical beam profile generated at the distal far field of the MCF to facilitate cell rotation. As demonstrated in Fig. 1a, the rotating elliptical beam profiles (Fig. 1b) are achieved through dynamic phase modulation employing the phase-only spatial light modulator (SLM) on the proximal side of the fiber. The computer-generated holograms are calculated by the previously proposed physics-informed deep neural network in a quasi-video rate, ensuring high-fidelity beam control in the MCF-OCR[41]. The cell rotation mechanism within the MCF-OCR is primarily driven by the unique characteristics of cells and the tailored elliptical beam profile.

As shown in Fig. 1c, cells typically have heterogeneous internal refractive index distribution. The asymmetric shape of the elliptical beam profile induces the optical gradient force on the cell. These forces exert a differential impact on the various regions of the cell due to its heterogeneous internal refractive index distribution. Thus, the cell is driven to align itself with the rotation of the elliptical beam profile. As the elliptical beam rotates, the cell trapped within the MCF-OCR follows its motion by the gradient force. Meanwhile, the cell remains stably trapped during the rotation due to the counter-balancing of the scattering forces exerted by the two laser beams, shown in Fig. 1d. The system cleverly employs an MCF and a strategically placed opposing single-mode fiber (SMF). The design of the rotating elliptical beam, with its near-Gaussian distribution, ensures the scattering forces from the two laser beams balance each other, minimizing the unwanted vibration during the cell rotation. The integration of the MCF-OCR into a lab-on-a-chip system is depicted in Fig. 1e. This system features a microcapillary, which is positioned between the two fibers and serves as a conduit for cell delivery to the optical manipulation region, facilitated by a controlled microflow. Once the cell is trapped, the microflow is terminated to ensure stability during the imaging process. The compact design of the lab-on-a-chip system enables smooth integration into a brightfield microscope, ultimately forming a comprehensive and efficient OPT system demonstrated in Fig. 1f. As depicted in Fig. 1g, the camera mounted on the microscope records 2D projections of the cell throughout its optically controlled rotation. These projections are then utilized for full-angle tomographic reconstruction.

### Autonomous tomographic reconstruction powered by computer vision

Reconstructing the 3D intensity distribution of cells from 2D projections typically requires laborious pre-processing, and the precise determination of the rotation angles remains challenging. To overcome these difficulties, we have developed an autonomous tomographic reconstruction workflow, as depicted in Fig. 2. This innovative approach streamlines the process and enables rapid and robust reconstruction for full-angle tomography based on sample rotation.

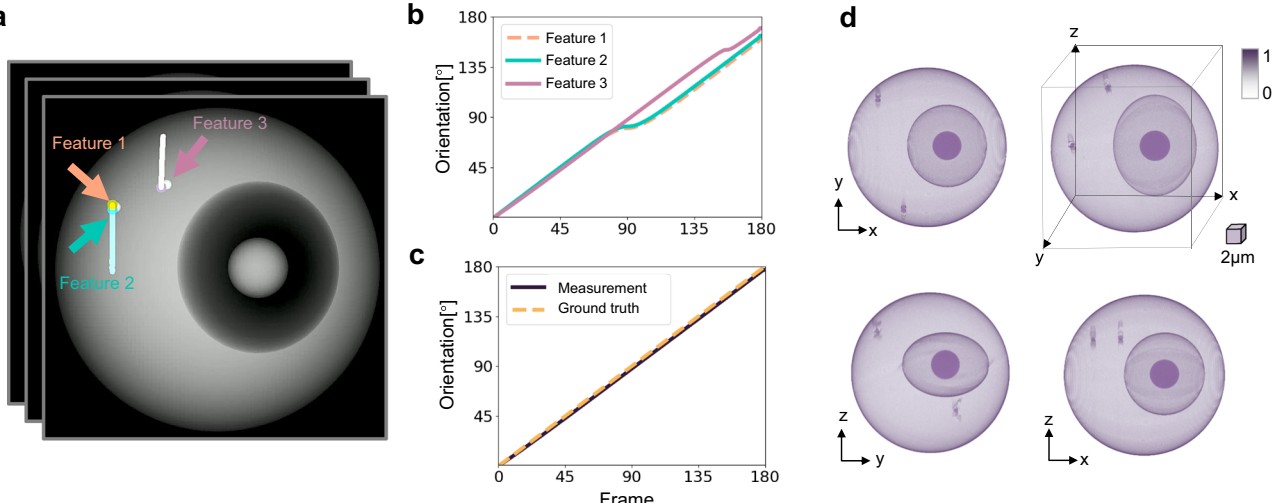

**Fig. 3 | Optical projection tomographic reconstruction of a cell phantom employing the proposed method. a** Tracking of the cell phantom rotation using optical flow. **b** The tracked rotation angle of the cell phantom. **c** Corrected rotation angles and the ground truth. **d** 3D intensity distribution reconstruction of the cell phantom. Scale cube $2 \times 2 \times 2 \, \mu m^3$. Source data are provided as a Source Data file.

The initial phase of our workflow employs the neural network YOLOv5[42], an object detection algorithm, to autonomously identify cells within raw microscopic images (Fig. 2a). Conventional training datasets for such CNNs do not encompass microscopic images of cells. However, the great generalization capability of YOLOv5 permits its application for cell detection within microscopic images employing transfer learning[43]. Here, we fine-tune the pre-trained YOLOv5 using a set of manually labeled microscopic cell images, optimizing the network specifically for this task. The efficiency of the tailored CNN is notable, achieving an average cell detection precision of 99.88% within our microscopic images. As demonstrated in Fig. 2b, subsequent to cell detection, the region of interest is extracted from the full-field images and cropped into smaller images of consistent dimensions. Utilizing the OpenCV video stabilization package[44], the cell's position within the cropped video frames is aligned, establishing a firm basis for the subsequent stages of our workflow. In order to achieve accurate image registration and 3D reconstruction with high contrast, it is crucial to accurately segment cells from images[45]. Deep learning has proven to be highly effective in achieving this, with a cell segmentation classifier used to differentiate cells from the background[46,47]. Similar to the method used for optimizing the object detection CNN, transfer learning is also employed for cell segmentation. Manually labeled images are implemented to fine-tune the trained classifier for precise cell segmentation in the proposed OPT system. To achieve optimal tomographic reconstruction, it is essential to precisely align the cell position in the cell rotation video[18]. Due to the high contrast of the images after cell segmentation, we extract the exact position of the cell in each frame. This is accomplished by determining the minimal enclosing circle of the cell's contour, a technique that has proven effective in various image analysis contexts[48]. As demonstrated in Fig. 2c, subsequent to the segmentation, we corrected the lateral movement of the cell by aligning it to the center of the frame, thus ensuring accurate image registration. Furthermore, accurate detection of the rotation angle is crucial for maintaining the fidelity of tomographic reconstructions[49]. Harris corner detector[50] is used to enable the extraction of dynamical features in the cell rotation videos. As shown in Fig. 2d, the motion of the extracted features is tracked and quantified by the optical flow[51–53]. The detected motion is then processed further to ascertain the rotation angle of the spherical cell. Leveraging the geometrical properties of the cell, the movement of these tracked features on the cell surface is translated into a quantifiable rotational angle. This conversion is pivotal in establishing the

precise orientation of the cell at each step, which is a prerequisite for high-quality, full-angle tomographic reconstruction. Once the precise orientations of the projections are determined, these data enable the construction of a sinogram, a 2D representation of the cross-sections of the cell captured at various angles. Each line of the sinogram corresponds to a specific projection angle, thus encapsulating the angular perspective of the subcellular structure. Ultimately, the tomographic reconstruction algorithm[54] is employed for the reconstruction of the 3D intensity distribution of the cell, allowing us to obtain a detailed 3D intensity distribution reconstruction of the cell, thereby contributing to a more in-depth understanding of its internal structures and properties shown in Fig. 2e.

## 3D isotropic reconstruction of cell phantoms

We have introduced the AI-driven autonomous workflow for accurate and rapid cell tomographic reconstruction. Accurate detection of the rotation angle is crucial for maintaining the fidelity of tomographic reconstructions[49]. Although the automatic tracking of cell orientation in microscopic images is highly desired, it remains a challenging task. Firstly, the complex and heterogeneous internal structure of the cellular structures might be indistinct or even invisible due to limitations in the resolution of optical microscopy and the natural transparency of cells. Also, during the rotation, different parts of the cell come into and go out of the field of view, making consistent tracking difficult. Furthermore, a gold standard method for precisely measuring the cell rotation angle is yet to be established. To address this challenge and validate the performance of the optical flow in tracking cell rotation, we have simulated a cell phantom, demonstrated in Supplementary Movie 2, which serves as a reliable reference for evaluating the accuracy and effectiveness of our proposed method.

In our simulation of full-angle OPT predicated on cell rotation, we smoothly rotate the 3D cell phantom through 180°, generating corresponding 180 projections, which are recorded as frames in the cell phantom rotation video. As exhibited in Fig. 3a, three dynamic features which are specifically corners with notable intensity changes on the cell surface, are extracted. These selected features undergo tracking for their lateral displacement using the optical flow technique. The rotation angle of the cell is subsequently computed from the tracked lateral movements of these features, as demonstrated in Fig. 3b. It can be noticed that the tracked rotation angle enters a phase of stagnation when the tracked feature moves from the front side to the back side of the projection. This phenomenon is attributable to the pixelation

effect of the images. When a feature on the cell moves across these pixels, even a small lateral shift of a few pixels can represent a significant change in the physical position on the cell surface. This becomes particularly noticeable at the edges of the cell, where the curvature of the cell body means that a small lateral shift in pixels corresponds to a large change in the orientation of the cell, in this case, approximately a 20° rotation. To circumvent this tracking error, we employ a multi-feature tracking approach and switch the tracked feature to the most optimal one when the rotation angle experiences stagnation. Figure 3c shows the corrected tracking result of the rotation. The mean measurement error of the tracked rotation angle is 1.49°, reflecting the high precision of the cell rotation tracking based on the multi-feature optical flow tracking approach. Once the precise rotation angle of each projection is obtained, the sinogram of the cell phantom can be generated. Utilizing the inverse Radon transform, cross-sections of the cell phantom at different depths are reconstructed from the angle-specific projections. The 3D intensity distribution of the cell phantom can then be reconstructed by stacking these cross-sections with spatial filtering. Figure 3d presents the 3D isotropic intensity distribution reconstruction of the cell phantom viewed from multiple perspectives. This demonstrates the capability of the approach to capture the full 3D structure of the cell phantom, providing a detailed and comprehensive representation that would be impossible to achieve with traditional 2D imaging techniques.

In order to quantitatively assess the fidelity of the proposed autonomous tomographic reconstruction workflow, we compare it with both the original cell phantom (Fig. 4a) and the conventional illumination scanning tomographic reconstruction (Fig. 4c) that only utilizes limited projection angles. Due to the physical constraints of the numerical aperture in microscope objectives, state-of-the-art illumination scanning tomography approaches cannot exceed an equivalent scanning angle range of 160° [4,6]. The optimal tomographic reconstruction of the cell phantom, achieved using this conventional method with limited angle coverage, is depicted in Fig. 4d, presenting cross sections at different planes. The proposed tomographic reconstruction method demonstrates significant improvement in axial resolution (along the $z$-axis), as illustrated in Fig. 4f, presenting reconstructed cross-sections at the same planes. To further validate the proposed tomographic reconstruction method, we conducted quantitative comparisons of the intensity distribution along the identified dashed lines in the cross-sectional tomographic reconstructions, as shown in Fig. 4g, h. The intensity distribution of the conventional illumination scanning tomography reconstruction, represented by the red curves, aligns with the red dashed line in Fig. 4d. Although this distribution generally correlates with the ground truth intensity distribution of the cell phantom, it deviates significantly at the cell phantom interfaces, indicating imprecise density values. On the other hand, the intensity distribution from the proposed cell rotation tomography reconstruction (represented by the blue curve in Fig. 4g, h) corresponds accurately with the ground truth intensity distribution of the cell phantom. The consistency of these distributions, which align along the blue dashed line in the cross sections (Fig. 4f), demonstrates the improved accuracy and superior reconstruction capability of our proposed methodology.

Moreover, to quantify the measurement error of the 3D reconstruction using both tomographic reconstruction approaches, we applied statistical error metrics: mean squared error (MSE), mean absolute error (MAE), and root mean square error (RMSE)[55]. We compared the tomographically reconstructed 3D volume with the original cell phantom. As shown in Table 1, the 3D reconstruction error significantly decreases when using our proposed tomographic reconstruction approach compared to the limited-angle illumination scanning tomography. Specifically, the MSE decreases by 66.7%, MAE by 58.0%, and RMSE by 43.8%. To provide a comprehensive evaluation of the accuracy of the 3D reconstruction, we incorporated the use of

multiscale structural similarity (MM-SSIM) and peak signal-to-noise ratio (PSNR) as assessment metrics. The MM-SSIM, a typical method used for quantifying image similarities[56], and the PSNR, a common benchmark for measuring the quality of reconstruction or compression[57], were calculated for the 3D reconstructions. The results, as detailed in Table 1, demonstrate a significant enhancement in the accuracy of 3D reconstruction when our cell-rotation-based tomography approach is utilized. Furthermore, there was a 7.1% increase in the MM-SSIM, indicating a closer alignment to the structure of the original cell phantom. Additionally, the PSNR, a measure of the fidelity of the reconstructed volume to the original, exhibited an improvement of 18.2% compared to the conventional tomography approach. Therefore, the proposed optical cell rotation tomography offers superior performance in terms of accurate tomographic reconstruction. Its efficacy surpasses the optimal results achievable through conventional illumination scanning tomography. By employing a rotational mechanism for cell imaging and incorporating advanced computational techniques, our approach is able to capture and reconstruct the 3D subcellular structure with remarkable precision and fidelity.

### 3D isotropic reconstruction of live HL60 human cancer cells

The accurate tomographic reconstruction of the cell phantom confirms the effectiveness of the proposed autonomous workflow for MCF-OCR-based OPT reconstruction. As a result, this approach can be extended and applied to reconstruct experimentally measured cell rotations, offering a powerful tool for analyzing cellular structures in detail. In order to perform full-angle OPT on live HL60 human leukemia cells, the cell is precisely rotated within the MCF-OCR system, generating 251 projections at orientations from 0° to 180° using the brightfield microscope. The subsequent phase involves processing these projections through the AI-driven autonomous tomographic reconstruction workflow that we have proposed. This stage begins with a pre-processing step that makes use of a pre-trained object detection CNN and a specific cell segmentation deep neural network. This technique enhances the data by refining and focusing on pertinent cell details. Following this initial refinement, the data undergoes a calibration process. This involves registering the minimal enclosing circle of the cell contour that has been detected, providing a more accurate dataset for further steps. With the refined and calibrated data, we move on to tracking multiple features on the cell simultaneously. For this, we employ the optical flow technique. This critical step allows us to compute the cell's rotation angle. We prioritize accuracy in this phase by ensuring the automatic selection of the optimal tracking feature. In the next step, the sinogram of the cell is generated and then translated into 2D cross-sections of the HL60 cell using the inverse Radon transform. The isotropically-resolved 3D intensity distribution of the live HL60 cell is demonstrated in Fig. 5 and Supplementary Movie 1. This reconstruction is made possible by stacking the processed 2D cross-sections and employing spatial filtering techniques.

The quality of the final 3D volumetric reconstruction is heavily influenced by the image pre-processing steps in the autonomous tomographic reconstruction workflow. As depicted in Fig. 6a, effective cell segmentation is crucial to eliminate background noise and enhance the clarity of the 3D volumetric reconstruction. However, as illustrated in Fig. 6b, even when the contrast of the 3D volumetric reconstruction is improved through cell segmentation, misalignment in the frames can still adversely affect the accuracy of the 3D reconstruction. The proposed AI-driven autonomous tomographic reconstruction workflow significantly improves the quality of the 3D intensity reconstruction of the human cancer cell, resulting in a robust and accurate tomographic reconstruction shown in Fig. 6c. Supplementary Movie 3 offers a rotational view of the 3D reconstruction of the HL60 cancer cell using different methods, providing a more in-depth comparison of the reconstruction results, and demonstrating

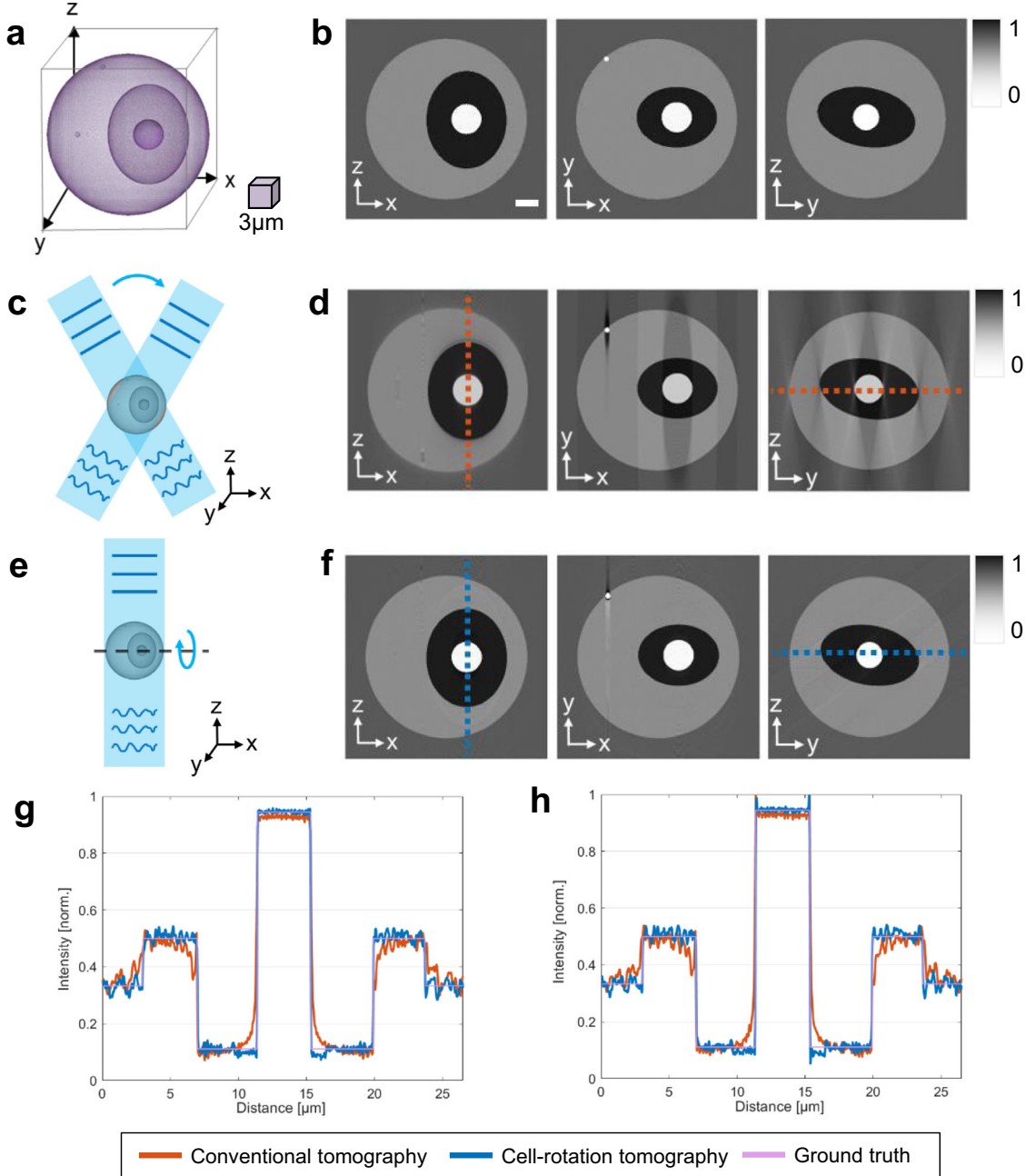

**Fig. 4 | Comparison of the optical projection tomographic reconstruction of the cell phantom using different methods. a** 3D visualization of the cell phantom. Scale cube $3 \times 3 \times 3\,\mu m^3$. **b** Cross sections of the cell phantom at XZ, XY, and YZ planes. Scale bar 3 μm. **c** Illustration of the conventional illumination scanning tomography principle applied to the cell phantom. **d** Cross sections of the reconstructed cell phantom using conventional illumination scanning tomography. **e** Illustration of the optical cell-rotation tomography. **f** Cross sections of the reconstructed cell phantom using the proposed autonomous cell-rotation tomography approach. **g, h** Quantitative comparison of the intensity profile along the color-marked dash lines in the reconstructed **g** XZ plane cross-section and **h** YZ plane cross-section. The red line is conventional illumination scanning tomography reconstruction; the blue line is optical cell-rotation tomographic reconstruction; the purple line is the ground truth intensity of the cell phantom. Source data are provided as a Source Data file.

**Table 1 | Quantitative comparison of the cell phantom reconstruction using conventional illumination scanning tomography and the proposed autonomous cell rotation tomography**

|  | MSE | MAE | RMSE | MS-SSIM | PSNR |
|---|---|---|---|---|---|
| Conventional tomography | 0.0018 | 0.0262 | 0.0420 | 0.8888 | 27.5299 |
| Proposed tomography | 0.0006 | 0.0110 | 0.0236 | 0.9519 | 32.5294 |

the isotropically-resolved 3D reconstruction using the AI-driven tomographic reconstruction workflow. This enables a precise and accurate representation of the subcellular structures, enhancing our understanding of their intricate morphology.

## Discussion

We have demonstrated an AI-driven fiber-optic cell rotation tomography system that offers powerful 3D single-cell imaging, providing unique advantages not collectively realized with existing methodologies. This system allows for the efficient and robust reconstruction of

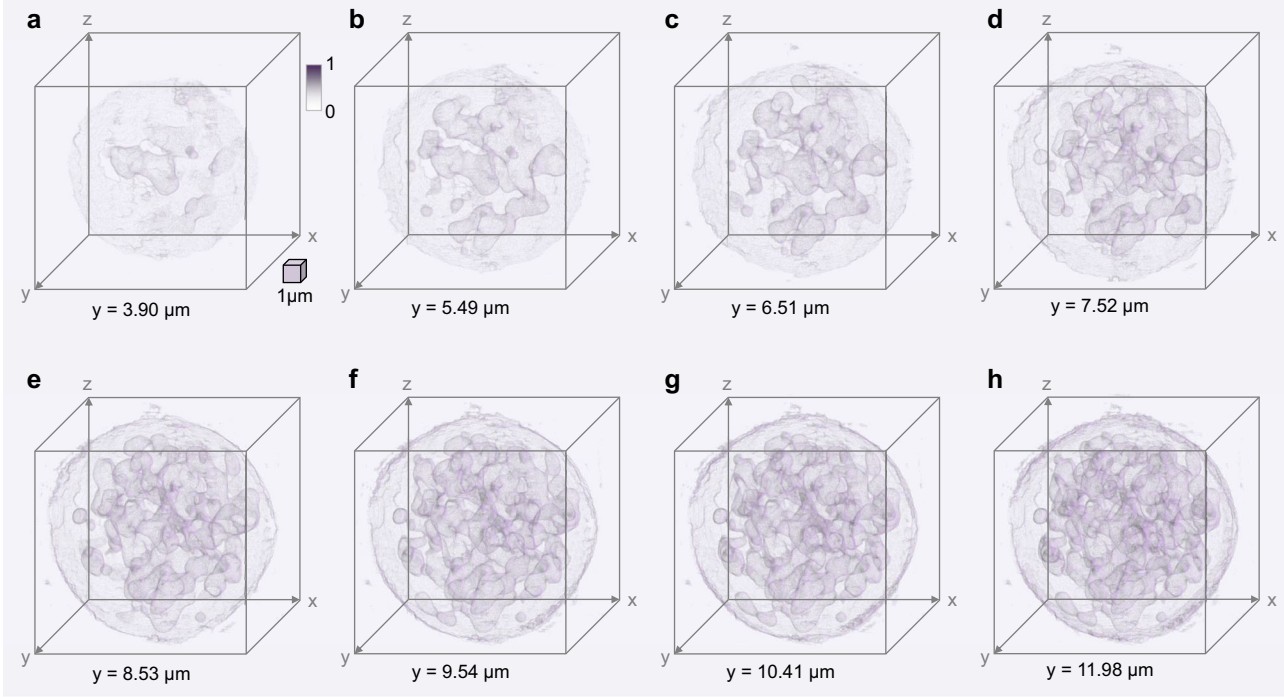

**Fig. 5 | Isotropically-resolved 3D intensity distribution reconstruction of a live HL60 human leukemia cancer cell.** The 3D reconstruction is sectioned along the y-axis with a depth of **a** 3.90 μm **b** 5.49 μm **c** 6.51 μm **d** 7.52 μm **e** 8.53 μm **f** 9.54 μm **g** 10.41 μm **h** 11.98 μm. Scale cube 1 × 1 × 1 μm³. Source data are provided as a Source Data file.

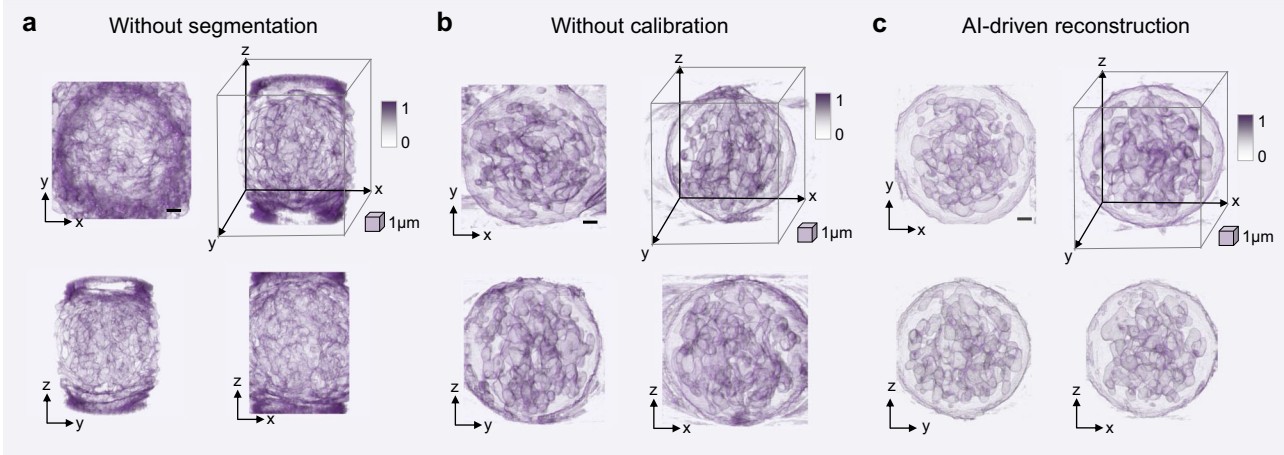

**Fig. 6 | Comparison of 3D intensity distribution reconstruction for a live HL60 human leukemia cancer cell rotated in the MCF-OCR. a** 3D reconstruction in the absence of cell segmentation. **b** Reconstruction with cell segmentation but lacking frame calibration. **c** Isotropic 3D reconstruction using the proposed AI-driven autonomous tomographic reconstruction workflow. Scale cube 1 × 1 × 1 μm³. Source data are provided as a Source Data file.

3D intensity distributions with diffraction-limited isotropic resolution. Notably, this 3D cell imaging performance can be attained using commercially available microscopes due to the compact design of our MCF-OCR lab-on-a-chip system. Furthermore, the autonomous tomographic reconstruction workflow substantially simplifies the full-angle tomography process, thereby increasing its accessibility for broader applications. Through the implementation of the autonomous tomographic reconstruction workflow on a cell phantom, we have affirmed the effectiveness of our tomographic reconstruction methodology. High-fidelity 3D intensity distribution reconstructions were realized for both the cell phantom and live human cancer cells, underscoring the precision of cell orientation in each projection and the overall

accuracy of our system. We have additionally integrated machine learning-empowered cell segmentation and frame calibration algorithms into our workflow, resulting in an optimal adjustment of experimentally captured images of optically rotated cells. This advancement considerably enhanced the robustness of our tomographic reconstruction process, reinforcing the transformative potential of our approach in the field of tomographic imaging.

Conventional cell tomography methods are often constrained by the range of scanning angle, which consequently leads to the missing cone problem[8]. In contrast, our system can solve this problem by enabling full-angle tomography, providing isotropic resolution in 3D. This is achieved by utilizing the MCF-OCR to execute precisely

controlled cell rotation. Although iterative reconstruction algorithms[8,9] and deep learning methodologies[12,13] have made remarkable advancements in enhancing the resolution of illumination scanning tomography, our proposed tomography approach can provide the much-needed ground truth measurement data for evaluating these algorithm-estimated tomographic reconstructions. Although the optical control of cell rotation is improved with the MCF-OCR when compared to few-mode fiber-optic cell rotation[28], rotation around a slightly tilted axis could potentially impact the fidelity of the tomographic reconstruction. By further implementing the adaptive tomographic control of the light field[26] through the MCF, we anticipate enhancing the robustness of the tomographic reconstruction for optimal optical manipulation.

The proposed tomography method introduces a general reconstruction workflow for tomography techniques based on cell rotation, offering more accurate 3D volumetric reconstruction compared to conventional limited-angle tomography. This autonomous tomography reconstruction workflow can be readily extended to various cell rotation tomography methods, including microflow cell rotation[17–19], dielectrophoretic cell rotation[21,22], acoustic microstreaming cell rotation[23,24], or combined acoustic and optical manipulation[25]. Moreover, we envision that the computer vision technologies presented here could have broad applicability across various tomography modalities. This could include their potential use in refractive index tomography[58,59], fluorescence tomography[18], or X-ray tomography[60]. Therefore, our work could not only advance the field of cell-rotation-based tomography but also open up new possibilities for the integration of computer vision technologies in diverse tomographic modalities.

## Methods
### Experimental setup
The comprehensive experimental setup of the proposed tomography system is demonstrated in Supplementary Fig. 2. This system employs an MCF (FIGH-350S, Fujikura) to dynamically modulate the output light field, controlled by an SLM (PLUTO-2, Holoeye), which further enables optical control of cell rotation. The core of the setup, the MCF-OCR, comprises the MCF and an opposing SMF (SM600, Thorlabs). The SMF has dual functions: it accommodates the reference beam used for calibrating the MCF, and it is also used as the waveguide for the near-infrared fiber laser (Eylsa 780; Quantel), which is responsible for optical trapping. The output laser power from each optical fiber is kept under 40 mW to minimize the risk of photodamage and photothermal effects. A custom-built brightfield microscope is used to image the cell rotation process. We use a blue light-emitting diode (LED) (M455L4, Thorlabs) as the light source for bright-field imaging. The area of optical manipulation is magnified by a microscope objective (50×, 0.42 NA, Mitutoyo) and tube lens (TTL200, Thorlabs), and this magnified image is projected onto a recording camera (Ueye CP, IDS) via lens systems. To achieve clear and high-contrast imaging, short-pass filters (FES0500, Thorlabs) are positioned before the camera to eliminate scattered light from the laser beams used for optical manipulation.

### In situ calibration of phase distortion in the MCF
Each individual fiber core in the MCF acts as a separate single-mode waveguide with the capacity to carry light from one end of the fiber to the other. However, inherent variations stemming from the manufacturing process and its physical characteristics can cause each fiber core to introduce unique phase alterations to the transmitted light. These differences in phase shifts among various fiber cores result in phase distortions across the output light field. To address this challenge, we utilize a specialized in-situ calibration method tailored for the MCF-OCR system. As depicted in Supplementary Figure 1, the laser source for the MCF is concurrently coupled into the SMF within the

MCF-OCR system. This beam travels through the MCF and interferes with the reference beam on the camera positioned on the MCF's opposite side. Subsequently, the phase distortion in the MCF is reconstructed from the off-axis hologram and compensated using the SLM. When the MCF-OCR system is integrated into a new microscope, further phase drifts, as well as temporal and bending phase distortions, can be in-situ calibrated through this approach. This method also allows us to compensate for bending-induced phase distortions, thereby significantly boosting the robustness of the MCF-OCR tomography system. Moreover, the latest advancements in our lab indicate that phase distortion in the MCF can be corrected using 3D-printed diffractive optical elements on the fiber facet[61,62]. These diffractive optical elements could potentially replace the SLM, resulting in a less costly, simpler, and more robust setup.

### Physics-informed neural network for fiber-optic manipulation
The generation of computer-generated holograms for complex wavefront shaping through MCFs has been a challenging task due to the random and discrete distribution of the fiber cores in the MCF. This complexity limits the real-time light field control in MCF-OCR. To overcome these challenges, a physics-informed neural network named CoreNet was developed[41]. CoreNet incorporates a diffraction model, a physical concept, into its network design. The incorporation of this physics-based model allows CoreNet to propagate the light field between the phase modulation plane and the target intensity plane numerically. This feature of CoreNet enables the network to efficiently search for the optimal phase modulation maps for the target image and also learn the mapping from the target images to the phase modulation holograms in an unsupervised manner. CoreNet is capable of generating tailored CGHs at a quasi-video rate of 7.1 frames per second, speeding up the computation time by two magnitudes compared to conventional algorithms. The application of CoreNet for generating tailored light fields in MCF-OCR has significant implications for real-time control of the light field, allowing for the accurate control of the optical force exerted on the cells during rotation. This enhanced control can lead to improvements in the quality of cell rotation tomography.

### Sample preparation
The HL60 is a myeloid precursor cell line, initially derived from an Acute Promyelocytic Leukemia (APL) female patient in 1977, and has been widely used in various research. This work utilizes the HL60/S4 line, a modified variant of the original HL60 lineage. The HL60/S4 cells, kindly provided by D. and A. Olins from the Department of Pharmaceutical Sciences, College of Pharmacy, University of New England, were rigorously authenticated prior to their application in our research[63]. The HL60 cells were cultured at 37 °C and 5% $CO_2$ in RPMI medium (Gibco) supplemented by 10% fetal bovine serum (FBS, Gibco) and 1% penicillin–streptomycin. After reaching the desired confluency, cells were centrifuged for 5 min at 115×$g$ and then resuspended in phosphate-buffered saline (PBS). For subsequent measurements, the cell suspension was diluted by adding PBS to rotate one cell at a time without flushing a second cell into the field of view. The diluted cell suspension was pumped into a specialized microchannel constructed from hollow square capillaries (CM Scientific), which have an inner diameter of 50 × 50 μm and a wall thickness of 25 μm.

### Inverse Radon transform for tomographic reconstruction
The inverse Radon transform is used for reconstructing 3D cell volume from 2D projections of the cell rotation. It operates under the principles of the Fourier slice theorem and the inverse Fourier transform. The Fourier slice theorem specifies that the Fourier transform of a Radon transform of an object, which is the sinogram, equates to a slice through the 3D Fourier transform of that object. Mathematically, this

can be written as:

$$\mathcal{F}[R(\theta, p)] = \mathcal{F}_{3D}(\omega\cos(\theta), \omega\sin(\theta), z) \tag{1}$$

where $\mathcal{F}[R(\theta, p)]$ is the Fourier transform of the sinogram $R(\theta, p)$ at a specific angle $\theta$ and position $p$. Meanwhile, $\mathcal{F}_{3D}(\omega\cos(\theta), \omega\sin(\theta), z)$ represents a slice through the 3D Fourier transform of the object at the same angle $\theta$ and frequency $\omega$, at a specified depth $z$ within the object. The 3D reconstruction is then performed by taking the inverse Fourier transform of the projections and summing over all angles. This process can be described by:

$$f(x, y, z) = \int \int \mathcal{F}^{-1}\{\mathcal{F}[R(\theta, p)]\} e^{i\omega(x\cos\theta + y\sin\theta)} \, dp \, d\theta \tag{2}$$

where $f(x, y, z)$ is the reconstructed 3D volume of the cell, $\mathcal{F}^{-1}\{\mathcal{F}[R(\theta, p)]\}$ is the inverse Fourier transform of the sinogram, $e^{i\omega(x\cos\theta + y\sin\theta)}$ is the Fourier kernel, which is used to transform the sinogram back into the spatial domain, and the integrals $dp$ and $d\theta$ are performed over the sinogram ($p$-axis) and all projection angles ($\theta$-axis), respectively.

### 3D visualization of the tomographic reconstruction

The 3D intensity distribution volume of the cell, which was reconstructed using the inverse Radon transform, was subsequently imported into ImageJ. Specifically, the data were processed using the Volume Viewer plugin[64]. A nearest-neighbor interpolation method was employed to optimize the visualization of the intensity distribution volume and to minimize sampling artifacts. This method, while computationally efficient, retains the original voxel values of the input dataset, thereby preventing the introduction of new, interpolated values that could potentially alter the original data. To further enhance the visualization and analysis of subcellular structures, the 2D gradient values of the volume were utilized as a filtering mechanism. These gradient values, indicative of the rate of intensity change across the image, are instrumental in identifying and delineating boundaries between distinct subcellular structures. By effectively harnessing these gradient values, we were able to generate a clearer, more detailed visualization of the internal architecture of cells, thereby facilitating a more nuanced understanding of their morphological characteristics.

### Statistics and reproducibility

We employed a specialized approach focusing on a single HL60 human cancer cell and a cell phantom to validate the proposed AI-driven tomographic reconstruction method. Due to the specific and exploratory nature of the research, traditional statistical analyses were not applicable, and the sample size, including one cell and one phantom, was chosen based on the objective of demonstrating the method's feasibility rather than for statistical generalization. Therefore, no statistical method was used to predetermine sample size. All data obtained from the experiments were included in the analysis, hence, no data were excluded from the analyses. The study's design did not incorporate randomization or blinding as it focused on the technical validation of the imaging method, therefore, the experiments were not randomized and the investigators were not blinded to allocation during experiments and outcome assessment. To ensure reproducibility, the source code[65] and test data[65] have been made publicly available, allowing the scientific community to replicate our findings and apply the methodology in similar settings.

### Data availability

The 3D reconstruction data generated in this study and the source data for the figures have been deposited in Figshare[66].

### Code availability

The source code for the AI-driven tomography reconstruction is publicly available on Github[65]. Pre-trained DNN for hologram generation is publicly available on Github.

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

## Acknowledgements

We appreciate the invaluable contributions made through discussions and assistance from Dr. Robert Kuschmierz, Elias Scharf, Jakob Dremel, Alexander Echeverría Kientzle, Nikolas Wohlgemuth, and Vireak Dam. Our special thanks go to Raimund Schlüßler for his commitment in culturing the cells. We would like to especially thank Paul Mueller for providing the raw data of the artificial cell phantom. We deeply appreciate the collegial support from all members of BIOLAS. We thank

the Czarske laboratory for critical feedback on the paper. The authors acknowledge financial support from the Shanghai Artificial Intelligence Laboratory, National Key R&D Program of China (2022ZD0160100), the National Natural Science Foundation of China (62376222), and Young Elite Scientists Sponsorship Program by CAST (2023QNRC001) (to J.S.), the German Research Foundation (DFG) under grant No. CZ55/40-1 (to J.S., N.K., and J.W.C.), and the Max Planck Society (to J.G.).

## Author contributions

J.S., N.K., and J.W.C. conceptualized the experiments. J.S. carried out the experiments. J.S. and B.Y. developed the algorithm and analyzed the results. J.S., N.K., and J.W.C. managed the research project. J.G., N.K., and J.W.C. supervised the project. J.S., N.K., and J.W.C. acquired the funding. J.S. and B.Y. prepared the paper. All authors reviewed the paper.

## Competing interests

The authors declare no competing interests.
