## [Peer Review File · Nature Communications]

REVIEWER COMMENTS

Reviewer #1 (Remarks to the Author):

Reviewer's report on the manuscript: AI-Driven Projection Tomography with Multicore Fibre-Optic Cell Rotation by Jiawei Sun et al.

The authors propose in their work the possibility to use Multi Core Fiberoptics (MCF) device for inducing cells' rotation. Such rotation can be exploited for probing the cell with a light beam from many different directions in order collect enough data to retrieve a projection tomograms of the same cell. The system based on MCF is intrnsecally a compact system and versitali tool for addressing dynamically and with enough speed the appropriate light beam spatial distribution for manipulte single cells. In fact, such cell rotator system allow to achieve full-angle projection tomography assuring isotropic resolution.

However, in my opinion what makes this work valuable is that the authors applied AI as tool for getting tomograms, i.e. they use AI for implementing the tomographic reconstruction by avoiding manual computational image processing that could become fully automatic. Authors furnished a validation by through the 3D reconstruction of a cell phantom and a real experiment with human cancer cells.

In my opinion this work is a very good and results deserved to be published.

However, some point should be addressed by the authors to make their manuscript to be suitable for publication.

Major issues:

- did the authors addressed the potential problem of phototoxicity in this light delivery system foprr cell manipulation?

- Usually in optical trapping systems the trapped objects oscillate and I guess this is the reason of image stabilization process in Fig. 2 b. Now in the supplementary file is reported that the overall MCF has a framerate of 7.1frames per second. How much oscillate the cells in the experiments. Maybe could be of interest of the readears to have the plot of position of the cell with time while the cell is rotated. Such plot of the tracking of one cell will help the reader. Thus, authors have to please include such tracking (related to the stabilization process) and explain how they operate the tracking.

- The cells oscillate only in the 2D or even in 3D? Please comments this point.

- What about any deformation of the cells du to optical force exerted on the cells during rotation?

Minor issues:

-Page 10: "CoreNet is capable of generating tailored CGHs at a quasi-video-rate, speeding up the computation time by two magnitudes compared to conventional algorithms. The application of CoreNet for generating tailored light fields in MCF-OCR has significant implications for real-time control of the light field, allowing for the accurate control of the optical force exerted on the cells during rotation.

Authros should give some numbers about the speed of the generation of CGH and the thus of light patterns variation. Which is the maximum rotation fot eh cells that can be achived?

- A couple of refs about deep-learning for cell tomography would be necessary to complete the references list, such as for example:

-

"Speeding up reconstruction of 3D tomograms in holographic flow cytometry via deep learning." Lab on a Chip 22.4 (2022): 793-804.

"Three-dimensional tomography of red blood cells using deep learning." *Advanced Photonics* 2.2 (2020): 026001-026001.

Provided that the authors will address the above points satisfactorily, the manuscript can be considered for publication.

Reviewer #2 (Remarks to the Author):

In this paper, the authors propose a novel approach that employs a compact multi-core fiber (MCF) based optic cell rotator system to facilitate precise optical manipulation of cells within a microfluidic chip. Moreover, the authors demonstrate an AI-driven tomographic reconstruction workflow. The performance of the proposed cell rotation tomography approach is validated through the three-dimensional reconstruction of a cell phantom and HL60 human cancer cells. Overall, this paper is generally convincing and straightforward. However, there are several problems that need to be addressed before its acceptance.

1. The authors use the inverse Radon transform to achieve three-dimensional tomography reconstruction. However, when imaging subcellular structures below 1 μ m in the 455nm visible light band, diffraction effects must be considered. Therefore, if the authors want to ensure the accuracy of their 3D tomographic reconstruction, they need to consider changing the inverse Radon transform to a diffraction tomography reconstruction algorithm.
2. The core development of this paper is using a compact MCF to rotate cells by modulating the light field, so the fundamental difficulty of this method is concentrated on the precise three-dimensional rotation registration. Although the authors have detailed discussions on the Harris corner detector, corner point track method, and rotation angle conversion equations in the supplementary material, these are common registration methods based on traditional image processing algorithms, just like the most traditional inverse Radon transform. On the contrary, the author uses a relatively new deep learning method in automatic ROI area target recognition and cell segmentation. The accuracy of 3D tomographic reconstruction of the whole technology does not seem to depend on the so-called AI-driven tomographic reconstruction workflow. Because, in my opinion, for Harris corner detecting and corner point tracking, which require higher registration accuracy, do not use AI-driven reconstruction workflow after all.
3. When considering the diffraction effect, will the registration accuracy of the 3D registration algorithm used in the paper be affected? For example, after the tracked corner point passes through thick cells, the three-dimensional space position of the corner point may not be accurately tracked due to the multiple scattering effect. In response to these problems, has the author thought about using deep learning technology to propose some new registration methods?
4. When calculating the rotation angle, the author only calculated the rotation angles in the X and Y directions. But in my opinion, due to the irregular jitter of trapped cells, the author should still consider the rotation angle in the Z direction (although the rotation angle is relatively small) to ensure the reconstruction accuracy.
5. The authors should provide quantitative experiments on 3D registration accuracy and 3D reconstruction accuracy in real systems. It is recommended to conduct a real experiment on a known shape and anisotropic microsphere or microrod, and accurately check the real 3D registration accuracy and 3D reconstruction accuracy besides using a simulation system and a cell phantom.
6. The author's comparison of tomography methods based on traditional multi-angle illumination is not fair because the tomographic ability and reconstruction accuracy of tomography methods based on traditional multi-angle illumination depend on the numerical aperture of the objective lens and the maximum oblique illumination angle. The author should compare the three-

dimensional tomographic accuracy of a microscopic imaging system with an objective lens numerical aperture of not less than 0.9NA and a maximum oblique illumination angle of not less than 80 degrees in Fig. 4 and Tab. 1.

7. Due to the strong relevance to this work, the recent review on deep learning in optical metrology should be cited in this paper. Light Sci Appl 11, 39 (2022).
<https://doi.org/10.1038/s41377-022-00714-x>

Reviewer #3 (Remarks to the Author):

The authors have assembled an optical tomography system which can be implemented into a conventional wide-field optical microscope. The system comprises of a dual-beam optical trap realized by a multi-core fibre from one side and a single-mode fibre from the opposite side. The trapping wavefront is modulated using an SLM which allows rotation of the trapped object about an axis perpendicular to the projecting direction. Moreover the authors employ couple of AI-driven procedures for processing of the projections in order to automate the process and enhance the quality of the reconstructions.

As for the fibre-optic part I am reviewing:

The authors have previously published the trapping system in Sun et al., "Rapid computational cell-rotation around arbitrary axes in 3D with multi-core fiber," Biomed. Opt. Express 12, 3423-3437 (2021). This system is based on the multi-core fibre and the opposing single-mode fibre and allows trapping and rotation of circular particles with heterogeneous distribution of refractive index. They also show, how to track precisely the angle of rotation which is essential in high-quality tomographic reconstructions.

I see the novelty in this manuscript in 1. integration of the trapping module into a lab-on-chip system and 2. an in-situ calibration step of the multi-core fibre which is added after the basic calibration. Ad. 1: the technology of lab-on-chip development and fabrication is not described here and hence not a focus of this paper. Ad 2. the calibration procedure is described and the in-situ part corrects for additional phase drifts introduced by integration into the lab-on-chip or temporal and bending phase distortions.

The authors also mention a potential simplification of the setup by using 3D-printed diffractive elements placed on the fibre facet. "These elements could replace the SLM..." - it is not clear however how, using these static elements, the dynamic modulation of the beam in order to rotate the trapped objects would be realized.

Overall, the use of dual-beam trap for optical tomography is a valid concept but it was published already in the Biomed. Opt. Express 12, 2021. Therefore I don't see the novelty in the published work in the fibre-based technology part.

The novelty of this paper lies in the AI-driven tomographic reconstruction which is applied in this fibre-based system but is potentially relevant also for other optical tomography systems. After all, this is also what the title suggests but should be better distinguished in the main text.

Couple of corrections:

Fig. S2 caption:

BE, beam expander's' - I see only one

CAM1 - proximal camera's' - I see only one

BB - description missing

semicolon missing before SMF1-2

Supplementary material:

C. Precise stabilization using minimal enclosing circle

'Then the centroid of the segmented cell, which represented by the centre point of the minimal enclosing circle.' - this sentence does not make sense.

'The detected centre coordinators...' - possibly 'coordinates'?

RESPONSE TO REVIEWERS' COMMENTS

First of all, it is our pleasure to acknowledge the kind, constructive comments from all reviewers. You can find our revised manuscript: "*AI-Driven Projection Tomography with Multicore Fibre-Optic Cell Rotation*" and the updated supplementary materials in the submission system.

We numbered the reviewers' comments in the order they appeared and answered each comment separately in blue (marked with: **Response to X.X**). The location of modifications in the manuscript is given and the **modified parts of the manuscript are cited and marked in red**.

Reviewer #1 (Remarks to the Author):

Reviewer's report on the manuscript: AI-Driven Projection Tomography with Multicore Fibre-Optic Cell Rotation by Jiawei Sun et al.

The authors propose in their work the possibility to use Multi Core Fiber-optics (MCF) device for inducing cells' rotation. Such rotation can be exploited for probing the cell with a light beam from many different directions in order collect enough data to retrieve a projection tomograms of the same cell. The system based on MCF is intrnsecally a compact system and versitali tool for addressing dynamically and with enough speed the appropriate light beam spatial distribution for manipulate single cells. In fact, such cell rotator system allow to achieve full-angle projection tomography assuring isotropic resolution.

However, in my opinion what makes this work valuable is that the authors applied AI as tool for getting tomograms, i.e. they use AI for implementing the tomographic reconstruction by avoiding manual computational image processing that could become fully automatic. Authors furnished a validation by through the 3D reconstruction of a cell phantom and a real experiment with human cancer cells.

In my opinion this work is a very good and results deserved to be published. However, some point should be addressed by the authors to make their manuscript to be suitable for publication.

Authors' response:

We would like to extend our sincere gratitude for the comprehensive review and insightful comments on our manuscript from reviewer#1. We appreciate your recognition of the potential of our work and its contribution to the field.

We are particularly grateful that you highlighted the novelty of our work lies on the AI-driven tomographic reconstruction workflow. In contrast to conventional sample-rotation-based tomographic reconstruction approaches that are often time-consuming and labour-intensive, our innovative use of AI leads to an automated and intelligent tomographic reconstruction process. This not only enhances the system's efficiency and reliability but also positions our method as a versatile tool. Remarkably, our AI-driven tomographic reconstruction workflow showcases flexibility in adapting to a diverse range of sample-rotation-based optical tomography techniques and can be further implemented in other tomography modalities

based on cytometry, dielectrophoretic or acoustic manipulation enabled sample rotation. This integration of artificial intelligence was indeed a cornerstone in our research and its application promises a significant shift in the tomographic reconstruction based on cell rotation approaches.

Once again, we appreciate your acknowledgement of this critical aspect which underscores the significance of our contribution. You can find the point-by-point response to your comments below.

Major issues:

1.1 Did the authors address the potential problem of phototoxicity in this light delivery system for cell manipulation?

Response to 1.1:

Thank you for bringing up this critical aspect of phototoxicity in the context of our fibre-optic manipulation system.

In contrast to conventional optical tweezer systems which use sharply focused beam induced gradient forces to trap the cell, divergent or moderately focused (low numerical aperture) beams are used in our multi-core fibre-optic cell rotator (MCF-OCR). Hence, the scattering forces are the dominant optical force component to trap the cell. Due to a much larger area of laser illumination, the photon density applied to the cell is hundreds of times smaller than the optical tweezers [R1]. Moreover, the laser power from each optical fibre is below 40 mW to minimize further photothermal effects [R2-3]. Hence, our MCF-OCR has the unique advantage of lower risk of photodamage compared to conventional optical tweezer systems. Also, we have included the explanation of the phototoxicity in the “Materials and methods” section.

[R1] Thalhammer, G., Steiger, R., Bernet, S., & Ritsch-Marte, M. (2011). Optical macro-tweezers: trapping of highly motile micro-organisms. *Journal of Optics*, 13(4), 044024.

[R2] Kreysing, M., Ott, D., Schmidberger, M. J., Otto, O., Schürmann, M., Martín-Badosa, E., ... & Guck, J. (2014). Dynamic operation of optical fibres beyond the single-mode regime facilitates the orientation of biological cells. *Nature communications*, 5(1), 5481.

[R3] Sun, J., Koukourakis, N., Guck, J., & Czarke, J. W. (2021). Rapid computational cell-rotation around arbitrary axes in 3D with multi-core fiber. *Biomedical Optics Express*, 12(6), 3423-3437.

Modifications:

Page 9 - Section “Materials and methods” - Subsection “Experimental setup”

New sentence:

“The output laser power from each optical fibre is kept under 40 mW to minimize the risk of photodamage and photothermal effects.”

1.2 Usually in optical trapping systems the trapped objects oscillate and I guess this is the reason of image stabilization process in Fig. 2 b. Now in the supplementary file is reported that the overall MCF has a framerate of 7.1frames per second. How much oscillate the cells in the experiments. Maybe could be of interest of the readers to have the plot of position of the cell with time while the cell is rotated. Such plot of the tracking of one cell will help the reader. Thus, authors have to please include such tracking (related to the stabilization process) and explain how they operate the tracking.

Response to 1.2:

Thank you for your insightful comment and the suggestion to provide a clearer understanding of the cell oscillation during the fibre-optic manipulation process. Yes, the oscillation of the trapped objects is common in optical trapping systems. Different from trapped microparticles which normally oscillate at a relatively high frequency, we have observed in the experiments that the oscillation of biological cells normally has a lower frequency. This discrepancy can be attributed to the viscoelastic properties of cells. The viscoelastic nature of cells can act as a damping mechanism, thereby reducing high-frequency oscillations.

We have included a new supplementary figure “Figure S6” shown below to demonstrate the oscillation in the XY plane. You are correct that such oscillation of the cell was a challenge for optimal tomography reconstruction and indeed motivated the inclusion of the image stabilization process. For efficient and precise tracking of the cell position during rotation, we implemented a systematic approach. Since the cell remains in the trapping region during the rotation, a pre-trained object detection neural network is implemented to automatically detect the cell from the large field of view microscopic images (1920 x 1200 pixels). Thus, the region of interest (160 x 160 pixels) which contains the cell is cropped out for tomography reconstruction. This can reduce the data size and thus boost the computation speed. However, as demonstrated in Figure 6b, the tomographic reconstruction degrades due to the lateral oscillation of the cell. To overcome this problem, we use a two-stage method. First, the OpenCV video stabilization library is used for rough stabilization of the cell position [R4]. In this library, the key features (cellular structures in this case) are detected and matched across the frames of the cell rotation video. After matching, the frames are translated to align the position of these detected features. However, it was challenging to precisely extract the subcellular structures which are below the diffraction limit, which further limits the accuracy of the stabilization. Hence, as demonstrated in Figure 2c, we employ a combined approach of cell segmentation and the minimal enclosing circle method in the second stage. On the one hand, as shown in Figure 6, the cell segmentation method can improve the contrast of the tomographic reconstruction. On the other hand, due to the high precision provided by the deep neural network driven cell segmentation, the accurate position of the cell can be detected by calculating the minimal enclosing circle of the feature in frames after segmentation. The advantage of the minimal enclosing circle method is that no matter the shape or orientation of the cell, the circle provides a consistent geometric figure. As a result, the exact location of the cell is determined by the central coordinate of this circle. By

modifying the frames based on this central coordinate, we manage to centralize the position of the cell throughout the video, which results in a more finely tuned stabilization.

Thank you again for the invaluable suggestion. We have revised the manuscript accordingly and hope that the revised text and figure are now clear for readers.

[R4] Adam Spannbauer (2021). Python Video Stabilization with OpenCV. https://github.com/AdamSpannbauer/python_video_stab.

Modifications:

Supplementary information Page 8 Section 5.C

“To calibrate these offsets, we employ the minimal enclosing circle method. This identifies the smallest circle that completely encompasses the cell in each frame. Thus, the precise position of the cell is determined by the centre coordinate of the minimal enclosing circle. By adjusting the frames accordingly, we ensure the cell maintains a consistent position in the video, achieving refined stabilization as shown in Fig. S6.”

New figure:

Figure S6. Quantitative comparison of cell position in (a) X-axis (b) Y-axis before and after implementing the stabilization process.

1.3 The cells oscillate only in the 2D or even in 3D? Please comments this point.

Response to 1.3:

Thank you for inquiring about the dimensional oscillation of cells in our system. Our dual-beam trap system inherently ensures a high level of axial stability, greatly owing to the prevalence of scattering forces which function similarly to a firmly anchored BBQ stick that allows an item to rotate steadily with just a light twist. In our system, this “light twist” is facilitated by the gradient force, induced by the real-time modulated light field. Although the gradient force used in our system is relatively weaker than the dominant optical force – the scattering force, the gradient force can still induce the smooth rotation of the cell while sustaining stability, effectively reducing any axial turbulence.

On the other hand, there exists slight axial oscillation of the cell during rotation. Implementing holographic refocusing [R5] and piezo-actuated microscopic objective [R6] can compensate for the axial oscillation. The reasons for the oscillation in different axes vary, the lateral oscillations are mainly caused by the intensity fluctuation of the laser beams emitted from the optical fibres. Meanwhile, the axial position stability of the cell during the rotation depends on the stability of the beam profile generated from the multi-core fibre, which determines the gradient force for controlling the cell rotation. The gradient force stability is characterized by the full-width half-maximum (FWHM) of the rotating beam profile, and the mean deviation of the FWHM in the minor and major axis is limited to 2% and 5%. Furthermore, in experiments, the cell is always in focus with a static microscopic objective without a piezo actuator for axial refocusing, demonstrating the high stability of the cell in the axial direction. Therefore, the slight axial oscillation can be neglectable in the three-dimensional tomographic reconstruction.

[R5] Wang, Z., Bianco, V., Pirone, D., Memmolo, P., Villone, M. M., Maffettone, P. L., & Ferraro, P. (2021). Dehydration of plant cells shoves nuclei rotation allowing for 3D phase-contrast tomography. *Light: Science & Applications*, 10(1), 187.

[R6] Fauver, M., Seibel, E. J., Rahn, J. R., Meyer, M. G., Patten, F. W., Neumann, T., & Nelson, A. C. (2005). Three-dimensional imaging of single isolated cell nuclei using optical projection tomography. *Optics express*, 13(11), 4210-4223.

1.4 What about any deformation of the cells due to optical force exerted on the cells during rotation?

Response to 1.4:

Thank you for bringing up the concern regarding the potential cellular deformation due to optical forces. As shown in Figure R2, it has been experimentally demonstrated that the deformation of HL60 cells using the optical forces happens when total laser power is 1 W (0.5 W per fibre) in the dual-beam fibre-optic trap [R7]. Under a reduced total power of 0.3 W, no deformation of the HL60 cells was observed. In our work, we have limited the total laser power to 0.08 W (0.04 W per fibre), and this power level is significantly below the threshold for deformation in HL60 cells. Furthermore, no deformation of the HL60 cell is observed in our experiments.

On the other hand, it is important to note that the deformation threshold is cell-specific. Different cell types can have varying thresholds for such optical deformation due to differences in their size, cellular composition or structure, surrounding medium etc. These intrinsic properties influence the refractive index distribution and homogeneity of the cell, thereby affecting how it interacts with and responds to optical forces. For instance, red blood cells have a lower threshold for optical forces induced deformation due to the smaller size and homogeneous refractive index distribution [R8-9]. Therefore, as the reviewer mentioned, the laser power used in the fibre-optic manipulation system needs to be adjusted carefully for different types of cells to avoid deformation.

Figure R2. (a) HL60 cell trapped with a total laser power of 0.3 W in a dual-beam fibre-optic trap. (b) HL60 cell is deformed with a total laser power of 1 W in a dual-beam fibre-optic trap. Adapted from [R5].

[R7] Huster, C., Rekhade, D., Hausch, A., Ahmed, S., Hauck, N., Thiele, J., ... & Cojoc, G. (2020). Stretching and heating cells with light—nonlinear photothermal cell rheology. *New Journal of Physics*, 22(8), 085003.

[R8] Bareil, P. B., Sheng, Y., Chen, Y. Q., & Chiou, A. (2007). Calculation of spherical red blood cell deformation in a dual-beam optical stretcher. *Optics Express*, 15(24), 16029-16034.

[R9] Guck, J., Ananthakrishnan, R., Mahmood, H., Moon, T. J., Cunningham, C. C., & Käs, J. (2001). The optical stretcher: a novel laser tool to micromanipulate cells. *Biophysical journal*, 81(2), 767-784.

Minor issues:

1.5 Page 10: “CoreNet is capable of generating tailored CGHs at a quasi-video-rate, speeding up the computation time by two magnitudes compared to conventional algorithms. The application of CoreNet for generating tailored light fields in MCF-OCR has significant implications for real-time control of the light field, allowing for the accurate control of the optical force exerted on the cells during rotation.” Authors should give some numbers about the speed of the generation of CGH and the thus of light patterns variation. Which is the maximum rotation for the cells that can be achieved?

Response to 1.5:

Thank you for highlighting the need for more quantitative details regarding the speed of the tailored modulation hologram generation using deep learning. As mentioned in the supplementary information on Page 5, the generation speed of the tailored full HD holograms (1920×1080 pixels) is 7.1 frames per second and is on-the-fly displayed on the spatial light modulator for controlling light field for optical manipulation. We appreciate your emphasis on this point and we have included this information in the main text as well.

Regarding the maximum rotation speed of the cells, the current rotation speed used for tomographic reconstruction is about 3 degrees per second and the hologram generation speed is not the limiting factor for reaching higher cell rotation speed. We found out in the experiments that the cell could no longer follow the rotation of the light field when the light field modulation speed increased to 7 frames per second. This is because as the cell rotates in the medium, it experiences a resistive torque due to the viscous forces acting on it. This torque, often referred to as the kinetic friction torque $\vec{\tau}_f$, and it increases linearly with the angular rotation speed ω according to the Stokes' law:

$$\vec{\tau}_f = \pi\eta D^3\omega$$

where η is the viscosity of the medium, and D is the diameter of the trapped cell. Therefore, higher optical torque is required to further increase the rotation speed of the cell. Increasing the laser power can be an option, but the laser power should be kept under the threshold of cell deformation and photodamage. In summary, although our system can generate holograms rapidly, the physical constraints from the frictional torque and the "laser safety" for cells limit the cell rotation speed.

Modifications:

Page 10 – Section “Materials and methods” – Subsection “Physics-informed neural network for fibre-optic manipulation”

“CoreNet is capable of generating tailored CGHs at a quasi-video-rate of 7.1 frames per second, speeding up the computation time by two magnitudes compared to conventional algorithms.”

1.6 A couple of refs about deep-learning for cell tomography would be necessary to complete the references list, such as for example:

-

"Speeding up reconstruction of 3D tomograms in holographic flow cytometry via deep learning." Lab on a Chip 22.4 (2022): 793-804.

"Three-dimensional tomography of red blood cells using deep learning." Advanced Photonics 2.2 (2020): 026001-026001.

Response to 1.6:

We appreciate your insightful recommendation on enhancing the references list. The first recommended paper entitled “Speeding up reconstruction of 3D tomograms in holographic flow cytometry via deep learning” introduced a novel multi-scale fully convolutional context aggregation network for rapid holographic reconstruction for 3D diffraction tomography based on flow cytometry. This approach maintains data quality with the network's compact design and is suitable for lab-on-a-chip devices with limited hardware resources. Moreover, our work has the potential to be further implemented in the flow cytometry based tomography and assisting the pre-processing of the holograms. The other recommended paper entitled “Three-dimensional tomography of red blood cells using deep learning”

introduces a new deep neural network for reconstructing 3D refractive index distributions from distorted 2D measurements to improve the 3D resolution from the illumination scanning diffraction tomography.

We have included both fascinating works in our reference list. We believe that the addition of these exciting works related to deep learning in 3D tomography would offer readers a more comprehensive understanding of the current advancements in this field.

Modifications:

Page 2 – Section “Introduction”

New references:

“These advancements have led to paradigm shifts in microscopy [33-35], including super-resolution, cell segmentation, and virtual staining.”

33. Daniele Pirone, Daniele Sirico, Lisa Miccio, Vittorio Bianco, Martina Mugnano, Pietro Ferraro and Pasquale Memmolo (2022). Speeding up reconstruction of 3D tomograms in holographic flow cytometry via deep learning. *Lab on a Chip*, 22, 793–804

34. Joowon Lim, Ahmed B. Ayoub, Demetri Psaltis (2020). Three-dimensional tomography of red blood cells using deep learning. *Advanced Photonics*, 2, 026001–026001

35. Yair Rivenson, Zoltán Göröcs, Harun Günaydin, Yibo Zhang, Hongda Wang, and Aydogan Ozcan (2017). Deep learning microscopy. *Optica*, 4(11), 1437-1443.

Provided that the authors will address the above points satisfactorily, the manuscript can be considered for publication.

Authors' response:

Thank you once again for your constructive comments. We greatly appreciate the time and expertise you've dedicated to reviewing our work. We have addressed all the mentioned points. We believe these revisions improved our manuscript, making it more comprehensive. We look forward to your positive response and are hopeful that our manuscript will meet the standards for publication.

Reviewer #2 (Remarks to the Author):

In this paper, the authors propose a novel approach that employs a compact multi-core fiber (MCF) based optic cell rotator system to facilitate precise optical manipulation of cells within a microfluidic chip. Moreover, the authors demonstrate an AI-driven tomographic reconstruction workflow. The performance of the proposed cell rotation tomography approach is validated through the three-dimensional reconstruction of a cell phantom and HL60 human cancer cells. Overall, this paper is generally convincing and straightforward. However, there are several problems that need to be addressed before its acceptance.

Authors' response:

We deeply appreciate the positive remarks from the reviewer#2. We're encouraged by your recognition of the novelty and potential impact of our multi-core fibre-based optic cell rotator system and the integration of an AI-driven tomographic reconstruction workflow. We acknowledge the concerns you've raised and are committed to addressing these issues to improve the clarity of our manuscript. Your insightful feedback is essential in helping us improve our paper to meet the standards required for *Nature Communications*.

2.1 The authors use the inverse Radon transform to achieve three-dimensional tomography reconstruction. However, when imaging subcellular structures below 1 μ m in the 455nm visible light band, diffraction effects must be considered. Therefore, if the authors want to ensure the accuracy of their 3D tomographic reconstruction, they need to consider changing the inverse Radon transform to a diffraction tomography reconstruction algorithm.

Response to 2.1

Thank you for your insightful comment. We acknowledge the importance of considering diffraction effects for increasing the reconstruction accuracy in optical tomography. However, our approach is primarily designed to be integrated with conventional bright-field microscopes, which typically utilize incoherent light sources. Also, an LED chip, which has low temporal and spatial coherence, is used as the imaging light source in this work. Therefore, the diffraction effects are substantially mitigated when compared to coherent imaging systems. Furthermore, the inverse Radon transform offers a good balance between image quality and computational efficiency [R10], which is crucial for improving the efficiency of the reconstruction workflow. We thank the reviewer's suggestion and recognize the potential advantages of optical diffraction tomography reconstruction algorithms such as Rytov approximation. Nevertheless, as the title suggests, this paper focuses on AI-driven projection tomography, and we believe it would be more relevant for future work focusing on optical diffraction tomography systems based on the MCF-OCR.

[R10] Müller, P., Schürmann, M., & Guck, J. (2015). ODTbrain: a Python library for full-view, dense diffraction tomography. *BMC bioinformatics*, 16, 1-9.

2.2 The core development of this paper is using a compact MCF to rotate cells by modulating the light field, so the fundamental difficulty of this method is concentrated on the precise three-dimensional rotation registration. Although the authors have detailed discussions on the Harris corner detector, corner point track method, and rotation angle conversion equations in the supplementary material, these are common registration methods based on traditional image processing algorithms, just like the most traditional inverse Radon transform. On the contrary, the author uses a relatively new deep learning method in automatic ROI area target recognition and cell segmentation. The accuracy of 3D tomographic reconstruction of the whole technology does not seem to depend on the so-called AI-driven tomographic reconstruction workflow. Because, in my opinion, for Harris corner detecting and corner point tracking, which require higher registration accuracy, do not use AI-driven reconstruction workflow after all.

Response to 2.2

Thank you for your insightful comments. We appreciate your recognition of the novelty of using a compact multi-core fibre-optic cell rotation system with dynamic light field modulation. We would like to clarify that "AI-driven reconstruction workflow" does not necessarily mean a complete reliance on machine learning technologies at every stage of the process. Instead, it reflects the strategic integration of artificial intelligence where they have clear advantages, while continuing to utilize traditional approaches where they are still effective.

In this work, we implemented an object detection convolutional neural network to achieve efficient and high-throughput cell detection. This not only expedites the processing speed but significantly reduces the data size, making subsequent processes more efficient. Additionally, the cell segmentation deep neural network enhances the precision of the cell's location detection. Also, as shown in Figure 5, it contributes significantly to the accuracy of rotation angle determination, resulting in an optimal 3D reconstruction. Therefore, the AI-driven workflow enhances both the efficiency and quality of the tomography reconstruction.

While a fully AI-integrated approach, such as an end-to-end deep neural network, could theoretically be applied for tomography. However, concerns about the reliability and explainability of such a fully AI-based method may arise. Thus, our current synergistic approach, combining the robustness of classical methods with the efficiencies of machine learning and computer vision, ensures both clarity and accuracy in the tomographic reconstruction process. We believe that this balanced methodology offers optimal results by harnessing the strengths of both AI and classical methods.

2.3 When considering the diffraction effect, will the registration accuracy of the 3D registration algorithm used in the paper be affected? For example, after the tracked corner point passes through thick cells, the three-dimensional space position of the corner point may not be accurately tracked due to the multiple scattering effect. In response to these problems, has the author thought about using deep learning technology to propose some new registration methods?

Response to 2.3

Thank you for raising crucial concerns regarding the 3D registration accuracy. You are correct in identifying the challenges on the tracked features which can be distorted due to scattering. In our current method, we make use of the Lucas-Kanade sparse optical flow technique. Instead of relying on exact feature points, this method employs a small window around each feature point. This small window strategy assists in counteracting minor inaccuracies and disturbances, as the algorithm considers the motion information of surrounding pixels which is a semi-global motion tracking. This ensures more consistent and robust tracking, especially in our case where single-pixel features might be distorted due to scattering or diffraction effects.

Moreover, as mentioned in the manuscript, we have implemented an object detection neural network and cell segmentation deep neural network to assist the image registration. Also, we have considered using deep learning for precise tracking of the cell rotation, for instance, deep learning based optical flow is a very interesting topic and could provide accurate tracking performance [R11-12]. However, a significant challenge is acquiring labelled datasets of cells, particularly for thick samples with scattering effects. Moreover, generating synthetic data that accurately simulates real-world cell dynamics and scattering effects is non-trivial, but this can be a very interesting topic for our future research. We greatly appreciate this suggestion and will consider these aspects in our future research.

[R11] Ilg, E., Mayer, N., Saikia, T., Keuper, M., Dosovitskiy, A., & Brox, T. (2017). FlowNet 2.0: Evolution of optical flow estimation with deep networks. In Proceedings of the IEEE conference on computer vision and pattern recognition (pp. 2462-2470).

[R12] Dosovitskiy, A., Fischer, P., Ilg, E., Hausser, P., Hazirbas, C., Golkov, V., ... & Brox, T. (2015). FlowNet: Learning optical flow with convolutional networks. In Proceedings of the IEEE international conference on computer vision (pp. 2758-2766).

2.4 When calculating the rotation angle, the author only calculated the rotation angles in the X and Y directions. But in my opinion, due to the irregular jitter of trapped cells, the author should still consider the rotation angle in the Z direction (although the rotation angle is relatively small) to ensure the reconstruction accuracy.

Response to 2.4

Thank you for raising this concern. As mentioned in the response to comment 1.3, our setup primarily facilitates rotation in the XY plane, and any jitter or oscillation in the Z direction is minimal and does not significantly affect the overall tomographic reconstruction. As you also mentioned in this comment, the tilted rotation about the Z-axis is relatively small (mean deviation of 3.28°), such small errors can be averaged out over the large number of projections (251 projections 180°). We also tried the filtered back projection reconstruction algorithm for tomography reconstruction to compensate for the Z-axis rotation. We observed that the improvement in the reconstruction quality was limited but the computational speed was significantly impeded. Given the relatively minimal gain in terms of reconstruction quality against the computational efficiency and one of our goals is to make the tomography workflow

accessible and usable in various application scenarios, we decided that the current workflow with balanced computation effort and accuracy was more suitable for our purpose.

2.5 The authors should provide quantitative experiments on 3D registration accuracy and 3D reconstruction accuracy in real systems. It is recommended to conduct a real experiment on a known shape and anisotropic microsphere or microrod, and accurately check the real 3D registration accuracy and 3D reconstruction accuracy besides using a simulation system and a cell phantom.

Response to 2.5

Thank you for your insightful suggestion. We indeed recognize the importance of quantitatively validating the 3D reconstruction accuracy in real-world experimental systems. To this end, we have implemented the proposed tomography reconstruction workflow on an experimentally rotated artificial cell phantom, which is a hydrogel sphere with two silica beads embedded. The additional experimental results are demonstrated in supplementary information and also shown below. We believe this detailed experimental demonstration reinforces the robustness and accuracy of our approach.

Modifications:

New section in supplementary information:

6. Validation of the AI-driven tomographic reconstruction workflow using an artificial cell phantom

To validate the precision and robustness of our AI-driven tomographic reconstruction workflow, particularly when applied to real experimental data, we employed an artificial cell phantom. The artificial cell phantom was trapped in a dual-beam fibre-optic trap and rotated by the microfluid control. This serves not only as a validation to the accuracy of our approach but also establishes its versatility across different cell rotation tomography systems. The artificial cell phantom was manufactured using a transparent hydrogel bead and two silica microbeads embedded with a diameter of 2.9 μm and 2.5 μm respectively [16].

The 3D intensity distribution of this artificial cell phantom, as reconstructed through our AI-enhanced workflow, is demonstrated in Fig. S10. In the reconstructed 3D intensity distribution, both microbeads are distinctly discernible across the XY, YZ, and XZ planes, validating the high fidelity of our tomographic reconstruction process. Furthermore, careful observation of the hydrogel's boundary reveals subtle textural variations, indicative of regions where the boundary material has been displaced due to the presence of the embedded microbeads. These intricacies are especially obvious in Fig. 10c. Overall, the microbeads compared to the hydrogel is clearly visible and the structure of the phantom is well-resolved in the 3D reconstruction.

Figure S10. 3D intensity distribution reconstruction of the artificial cell phantom containing two microbeads. (a) Top-down view along the Z-axis, scale bar 3 μm . (b) Isometric view at an angle of 45° , scale cube $3 \times 3 \times 3 \mu\text{m}^3$. (c) Lateral view aligned with the X-axis. (d) Lateral view aligned with the Y-axis.

To conduct a quantitative assessment of the tomographic reconstruction across various 3D orientations, we present cross-sectional images of the tomograph in the XY, XZ, and YZ planes, as depicted in Fig. 11a-c. The intensity profiles of the two microbeads, illustrated in Fig. 11d, demonstrating a high contrast between the silica microbead and its hydrogel surroundings. Moreover, the resolution achieved in XY and XZ planes is closely aligned, indicating a near-isotropic resolution in the 3D space.

Figure S11. Cross sections of the reconstructed artificial cell phantom at (a) XY plane (b) XZ plane (c) YZ plane, scale bar 3 μm . (d) Intensity profile along the colour marked lines in XY plane (blue) and XZ plane (yellow).

2.6 The author's comparison of tomography methods based on traditional multi-angle illumination is not fair because the tomographic ability and reconstruction accuracy of tomography methods based on traditional multi-angle illumination depend on the numerical aperture of the objective lens and the maximum oblique illumination angle. The author should compare the three-dimensional tomographic accuracy of a microscopic imaging system with an objective lens numerical aperture of not less than 0.9NA and a maximum oblique illumination angle of not less than 80 degrees in Fig. 4 and Tab. 1.

Response to 2.6

We appreciate your constructive suggestion. We agree with the reviewer that the maximum oblique illumination angle is crucial for the accuracy of the tomography system and is often determined by the numerical aperture (NA) of the illumination system or the microscope objective. Given the relationship between the NA and the angle θ which represents half the maximum oblique illumination angle

$$NA = n \sin(\theta).$$

Here, n is the refractive index of the immersion medium (typically 1 in air and about 1.518 for immersion oils). The scanning range of 120° (0.87 NA in air) was used to provide a general perspective based on commonly used objectives. However, we acknowledge that using more advanced objectives with an NA of at least 0.9 would indeed maximize the tomographic potential of traditional methods, and the corresponding oblique illumination angle is around 64.1°. We believe in the future with the technology development of the microscope objectives, the maximum oblique illumination angle can reach up to 80°. Therefore, we have revised our simulation data and generated new results with 160° illumination scanning range. We aim to ensure a more comprehensive and fair evaluation of the two approaches in Fig. 4 and Tab. 1. This would undeniably provide readers with a clearer understanding of the advantages and limitations of our proposed method in relation to state-of-the-art traditional tomography techniques under optimal conditions. Again, we deeply appreciate your expertise and guidance.

Modifications:

Page 5 – Subsection “3D isotropic reconstruction of cell phantoms”

“Due to the physical constraints of the numerical aperture in microscope objectives, **state-of-the-art** illumination scanning tomography approaches cannot exceed an equivalent scanning angle range of 160°.”

Revised Figure 4:

Figure 4 caption:

“(g) XZ plane cross-section and (h) YZ plane cross-section. Red line: conventional illumination scanning tomography reconstruction; blue line: optical cell-rotation tomographic reconstruction; purple line: ground truth intensity of the cell phantom.”

Revised Table 1:

Table 1. Quantitative comparison of the cell phantom reconstruction using conventional illumination scanning tomography and the proposed autonomous cell rotation tomography

	MSE	MAE	RMSE	MS-SSIM	PSNR
Conventional tomography	0.0018	0.0262	0.0420	0.8888	27.5299
Proposed tomography	0.0006	0.0110	0.0236	0.9519	32.5294

Page7 - Subsection “3D isotropic reconstruction of cell phantoms”

“Specifically, the MSE decreases by 66.7%, MAE by 58.0%, and RMSE by 43.8%.”

“Furthermore, there was a 7.1% increase in the MM-SSIM, indicating a closer alignment to the structure of the original cell phantom. Additionally, the PSNR, a measure of the fidelity of the

reconstructed volume to the original, exhibited an improvement of 18.2% compared to the conventional tomography approach.”

2.7 Due to the strong relevance to this work, the recent review on deep learning in optical metrology should be cited in this paper. Light Sci Appl 11, 39 (2022). <https://doi.org/10.1038/s41377-022-00714-x>

Response to 2.7

Thank you for sharing the extraordinary review on deep learning in optical metrology. We have now incorporated the suggested citation into the manuscript to ensure our readers have a more comprehensive understanding of the current advancements in this field. We appreciate your suggestion, and we believe that referencing this review enhances the context and depth of our paper.

Modifications:

Page 2 – Section “Introduction”

New references:

“Recently, the development of artificial intelligence (AI) and computer vision has revolutionized various aspects of optical metrology [31,32].”

31. Chao Zuo, Jiaming Qian, Shijie Feng, Wei Yin, Yixuan Li, Pengfei Fan, Jing Han, Kemao Qian & Qian Chen (2022). Deep learning in optical metrology: a review. Light: Science & Applications, 11(1), 39.

32. Shijie Feng, Qian Chen, Guohua Gu, Tianyang Tao, Liang Zhang, Yan Hu, Wei Yin, Chao Zuo (2019). Fringe pattern analysis using deep learning. Advanced Photonics, 1(2): 025001-025001.

Reviewer #3 (Remarks to the Author):

The authors have assembled an optical tomography system which can be implemented into a conventional wide-field optical microscope. The system comprises of a dual-beam optical trap realized by a multi-core fibre from one side and a single-mode fibre from the opposite side. The trapping wavefront is modulated using an SLM which allows rotation of the trapped object about an axis perpendicular to the projecting direction. Moreover the authors employ couple of AI-driven procedures for processing of the projections in order to automate the process and enhance the quality of the reconstructions.

Authors' response:

Thank you for the comprehensive summary of our work. Your overview accurately captures one of the most important advantages of our tomography system – the capability of being integrated into conventional bright-field microscopes. Thus, in the hardware part, the optical system shows the potential for wide applications due to the easy integration with conventional microscopes. Furthermore, in the software part, our primary goal was to enhance both the quality and efficiency of the cell-rotation tomography reconstruction, automating the reconstruction process using machine learning and computer vision technologies. We believe the proposed method not only streamlines the reconstruction process in our multi-core fibre-optic cell rotation system but can also be implemented to other cell tomography systems. Thank you for your invaluable suggestions and corrections. We have addressed your comments and have revised our manuscript accordingly.

As for the fibre-optic part I am reviewing:

3.1 The authors have previously published the trapping system in Sun et al., "Rapid computational cell-rotation around arbitrary axes in 3D with multi-core fiber," Biomed. Opt. Express 12, 3423-3437 (2021). This system is based on the multi-core fibre and the opposing single-mode fibre and allows trapping and rotation of circular particles with heterogeneous distribution of refractive index. They also show, how to track precisely the angle of rotation which is essential in high-quality tomographic reconstructions.

Response to 3.1:

Thank you for your meticulous review and for pointing out our previous work published in Biomedical Optics Express. You are right in noting that the fundamental configuration of the multi-core fibre-optic manipulation system was introduced in that publication, and that configuration serves as the basis for the cell-rotation-based tomography employed in our current work. We also demonstrated a manually tracked rotation angle in the previous work, but it can be hardly described as "precisely" compared to the new AI-driven approach. It is tricky and time-consuming to align the images and track the angle manually, therefore, only 9 projections are tracked for evaluation for a 180° cell rotation. In contrast, 251 projections are aligned and tracked automatically in this work due to the massive improvement in efficiency by using the AI-driven pre-processing, enabling accurate tomography reconstruction.

3.2 I see the novelty in this manuscript in 1. integration of the trapping module into a lab-on-chip system and 2. an in-situ calibration step of the multi-core fibre which is added after the basic calibration. Ad. 1: the technology of lab-on-chip development and fabrication is not described here and hence not a focus of this paper. Ad 2. the calibration procedure is described and the in-situ part corrects for additional phase drifts introduced by integration into the lab-on-chip or temporal and bending phase distortions.

Response to 3.2:

Thank you for highlighting the areas of novelty of the fibre-optic manipulation system in our manuscript. Although the current manuscript mainly focuses on the AI-driven projection tomography reconstruction, we also demonstrate several advancements in the multi-core fibre-optic manipulation system to improve the robustness of the multi-core fibre-optic system. We agree with the reviewer's insights that the in situ calibration procedure of the multi-core fibre is the key improvement for the successful integration of the proposed fibre-optic manipulation system into lab-on-chip. Furthermore, we also implemented the physics-informed deep neural network for real-time hologram generation in the fibre-optic manipulation system. With these new methods embedded, the MCF-OCR becomes much easier to be integrated into conventional microscopes and facilitates the cell-rotation based tomography.

3.3 The authors also mention a potential simplification of the setup by using 3D-printed diffractive elements placed on the fibre facet. "These elements could replace the SLM..." - it is not clear however how, using these static elements, the dynamic modulation of the beam in order to rotate the trapped objects would be realized.

Response to 3.3:

Thank you for highlighting the potential ambiguities concerning our proposal to use 3D-printed diffractive optical elements to replace the spatial light modulator (SLM). You're correct that the SLM offers great flexibility by enabling the dynamic generation of any desired light field and this is exactly the core technology in our proposed multi-core fibre-optic cell rotation system. The primary motivation behind introducing diffractive optical elements is to simplify the optical setup, making the system more affordable and can thus be widely used. The printed diffractive optical elements can compensate for the intrinsic phase distortion in the multi-core fibre and generate a focused beam at the fibre output, potentially simplifying the multi-core fibre-based optical trap or stretcher.

However, when it comes to the dynamic modulation required for cell rotation, we do recognize the challenges posed by the static diffractive optical elements. Therefore, special tricks have to be implemented to control the cell rotation instead of a dynamically modulated light field. One possible approach might be the generation of vortex beams using static phase masks [R13]. Such beams could potentially exert the necessary torques for cell rotation by using the orbital angular momentum of light [R14]. We appreciate this feedback and will strive to clarify this aspect further in the manuscript, ensuring readers understand both the limitations and potential opportunities provided by static diffractive elements.

[R13] Arrizón, V., Ruiz, U., Sánchez-de-la-Llave, D., Mellado-Villaseñor, G., & Ostrovsky, A. S. (2015). Optimum generation of annular vortices using phase diffractive optical elements. *Optics letters*, 40(7), 1173-1176.

[R14] Simpson, N. B., Dholakia, K., Allen, L., & Padgett, M. J. (1997). Mechanical equivalence of spin and orbital angular momentum of light: an optical spanner. *Optics letters*, 22(1), 52-54.

Modifications:

Supplementary information: Page 3 – Section 2

These diffractive optical elements could potentially replace the SLM, paving the way for a less costly, simpler, and more robust setup. **Specifically, the previously reported works use the diffractive optical elements for calibrating the phase distortion and generating a focused beam at the MCF output. It remains challenging to create the dynamically modulated light field reported in this work. An alternative approach could be employing a static phase mask to generate a vortex beam for generating the required torque for rotating the cell in the MCF-OCR [4,5]. Also, at the present stage, the calibration quality achieved using diffractive optical elements can hardly reach similar quality compared to using SLMs due to the limited resolution of lithography. Consequently, for optimal calibration, we continue to employ SLMs in this work.**

3.4 Overall, the use of dual-beam trap for optical tomography is a valid concept but it was published already in the *Biomed. Opt. Express* 12, 2021. Therefore I don't see the novelty in the published work in the fibre-based technology part.

The novelty of this paper lies in the AI-driven tomographic reconstruction which is applied in this fibre-based system but is potentially relevant also for other optical tomography systems. After all, this is also what the title suggests but should be better distinguished in the main text.

Response to 3.4:

Thank you for taking the time to read our previously published paper and providing an insightful evaluation of the novelty of our manuscript. We acknowledge that the multi-core fibre-based dual-beam trap was first introduced in *Biomed. Opt. Express* 12, 2021. However, it's crucial to note that the system presented therein lacked the capability for optical tomography. This limitation arose from the manual rotation angle tracking process, which was both time-consuming and imprecise, as detailed in response to 3.1. Optical tomography based on the multi-core fibre-optic trap was not demonstrated in the previous paper and it finally becomes possible due to the AI-driven tomographic reconstruction workflow which is the highlight of this paper. We deeply appreciate that you also emphasized that the novelty of this paper lies in the AI-driven tomographic reconstruction. Thanks for your kind suggestion, we have made revisions to our manuscript to accentuate this distinction more clearly.

Modifications:

Page2 Section "Introduction"

“In this paper, we introduce an AI-driven optical projection tomography (OPT) system that utilizes the multi-core fibre-optic cell rotator (MCF-OCR). This innovation effectively bridges the existing gap between fibre-optic manipulation and optical tomography. The core of this system is an AI-driven autonomous tomography reconstruction workflow, powered by emerging computer vision technologies, enhancing the robustness and efficiency of optical tomography systems.”

Couple of corrections:

3.5 Fig. S2 caption:

BE, beam expander's' - I see only one

CAM1 - proximal camera's' - I see only one

BB - description missing semicolon missing before SMF1-2

Response to 3.5:

Thank you for pointing out these errors in our manuscript, and we sincerely apologize for the oversight. We have corrected the caption of Fig. S2 as shown below. We appreciate your correction, which helps improve the quality of our work.

Modifications:

Supplementary information: Page 3 – Figure S2

“Figure S2. Schematic representation of the comprehensive experimental setup for the MCF-OCR tomography system. BB, beam blocker; BE, beam expander; BS1-2, non-polarizing beamsplitters; C1-4, collimators; CAM1, proximal camera; CAM2, microscope camera; CL, condenser lens; HWP, half-wave plate; L1-5, lenses; LED, light-emitting diode; M, mirror; MO1-2, microscope objectives; ND, neutral density filters; PBS, polarizing beamsplitter; PL, polarizers; QWP, quarter-wave plate; SF, spatial filter; SLM, spatial light modulator; SMF1-2, single-mode fibres; SPF, short pass filter; TL, tube lens.”

3.6 Supplementary material: C. Precise stabilization using minimal enclosing circle 'Then the centroid of the segmented cell, which represented by the centre point of the minimal enclosing circle.' - this sentence does not make sense. 'The detected centre coordinators...' - possibly 'coordinates'?

Response to 3.6:

Thank you for pointing out the errors in our description. To clarify, what we intended to convey is that the position of the segmented cell is determined by the centre point of the minimal enclosing circle. Also, you are right that the “coordinators” should be “coordinates”. We appreciate your feedback and have revised the sentences in this paragraph accordingly to ensure clarity and corrected the wording in our manuscript.

Modifications:

Supplementary information: Page 8 – Section 5.C

“To calibrate these offsets, we employ the minimal enclosing circle method. This identifies the smallest circle that completely encompasses the cell in each frame. Thus, the precise position of the cell is determined by the centre coordinate of the minimal enclosing circle. By adjusting the frames accordingly, we ensure the cell maintains a consistent position in the video, achieving refined stabilization as shown in Fig. S6.”

REVIEWERS' COMMENTS

Reviewer #1 (Remarks to the Author):

The revised manuscript is suitable for publication. In fact, in my opinion the authors have afforded all the raised points.
The work is very good and deserves to be published.

Reviewer #2 (Remarks to the Author):

The authors have revised the article according to the comments one by one and all my concerns are addressed. This version is suitable for its publication on Nature Communications.

Reviewer #3 (Remarks to the Author):

The authors have answered all my concerns. Thank you. I recommend the manuscript for publication in its revised form.

RESPONSE TO REVIEWERS' COMMENTS

Reviewer #1 (Remarks to the Author):

The revised manuscript is suitable for publication. In fact, in my opinion the authors have afforded all the raised points.

The work is very good and deserves to be published.

Authors' response:

Thank you for confirming that your concerns have been resolved and for recommending our manuscript for publication. Your feedback has been invaluable.

Reviewer #2 (Remarks to the Author):

The authors have revised the article according to the comments one by one and all my concerns are addressed. This version is suitable for its publication on Nature Communications.

Authors' response:

We appreciate your acknowledgment of the addressed concerns and your recommendation for publication in Nature Communications. Thank you for your guidance.

Reviewer #3 (Remarks to the Author):

The authors have answered all my concerns. Thank you. I recommend the manuscript for publication in its revised form.

Authors' response:

Thank you for your positive assessment and support for publication. We are pleased our revisions met your expectations.